# Mechanosensor-mediated Hsp70 phosphorylation orchestrates the landscape of the heat shock response

Siddhi Omkar[1], Jake T. Kline[2], James H. Grissom [1], Diyun Sun [3], Richard J. Chi[1], Jared A. M. Bard[4], Luca Fornelli [2] & Andrew W. Truman [1] ✉

Cells must respond rapidly to heat stress by activating multiple signaling pathways that preserve proteostasis. In budding yeast, this includes induction of Hsf1 and Msn2/4-mediated transcription, cell integrity signaling, stress-triggered phase separation of proteins, and inhibition of translation. How these pathways are so rapidly activated and coordinated remains unclear. We show that the mechanosensor Mid2 senses heat-induced membrane stretch and leads to rapid phosphorylation of the cytosolic Hsp70 Ssa1 at a well-conserved threonine (T492). Phosphorylation of T492 leads to epichaperome rearrangement promoting fine-tuning of multiple cellular processes including translational pausing, HSF activity, MAPK signaling and stress granule resolution. Taken together, these results provide a comprehensive, unified theory of the global yeast heat shock response mediated by the Hsp70 chaperone code.

The heat shock protein 70 (Hsp70) family represents a well-conserved yet functionally diverse group crucial for various cellular housekeeping functions[1,2]. The Hsp70 proteins are integral to the Heat Shock Response (HSR) because they bind and stabilize unfolded or misfolded proteins, facilitating their refolding or degradation. The yeast cytosolic Hsp70 family consists of the Ssa1-4 isoforms, with Ssa1 and Ssa2 constitutively expressed and Ssa3 and Ssa4 stress-inducible[3]. Despite their high sequence similarity, these isoforms exhibit distinct expression patterns and functions under stress conditions[3]. Hsp70s are composed of three main domains: the N-terminal ATPase domain, the substrate-binding domain (SBD), and the C-terminal "lid". Hydrolysis of ATP to ADP in the N-terminal domain drives conformational changes that regulate substrate binding, folding and release. This allosteric mechanism is essential for the chaperone activity of Hsp70, allowing it to transiently bind to a wide array of client proteins, thereby preventing aggregation and assisting in proper folding[4,5]. Chaperones exist as part of a wider Environmental Stress Response (ESR), a highly conserved mechanism that allows cells to adapt to various stress conditions, including heat, osmotic shock, oxidative stress, and nutrient deprivation. Upon exposure to these stressors, yeast cells globally downregulate genes involved in growth and proliferation while upregulating a suite of stress-responsive genes[6]. These genes encode proteins that aid in detoxification, repair damaged macromolecules, and maintain cellular integrity[5]. The Heat Shock Response (HSR), an evolutionarily conserved protective mechanism from yeast to humans, promotes protein homeostasis and cellular fitness during proteotoxic stress. As a component of the broader environmental stress response (ESR), HSR operates through two primary transcription factors, Hsf1 and Msn2/4[7]. The heat shock response (HSR) is a subset of the ESR and is fundamental for survival at high temperatures. This response is characterized by the rapid induction of heat shock proteins (HSPs) that include Hsp70 and Hsp90[8,9]. The chaperone titration model suggests that Hsf1 is kept inactive during basal conditions through interactions with Hsp70. Upon heat shock, the accumulation of misfolded proteins titrates Hsp70 away from Hsf1, allowing Hsf1 to bind Heat Shock Elements (HSEs), initiating the transcription of HSP genes[10]. This model, however, does not explain the rapid Hsf1 activation observed in response to heat shock, which occurs faster than the time required for titration to occur. The interaction between Hsp70 and Hsf1 occurs via the SBD domain of Hsp70 and two

[1]Department of Biological Sciences, The University of North Carolina at Charlotte, Charlotte, NC, USA. [2]Department of Biology, University of Oklahoma, Norman, OK, USA. [3]Department of Biochemistry & Biophysics, Texas A&M University, College Station, TX, USA. [4]Department of Biology, Texas A&M University, College Station, TX, USA. ✉e-mail: atruman1@uncc.edu

sites on Hsf1, the CE2 site in the C-terminal activation domain (C-AD) and a recently identified site in the N-terminal activation domain (N-AD)[9,11]. Studies have shown that Hsf1 activation can be regulated by accelerating the substrate release kinetics of Hsp70 in the nuclear compartment[11]. Other proposed HSR mechanisms include the direct sensing of heat by Hsf1[12], hyperphosphorylation[13], and condensation of Hsf1[10,14].

In parallel with the HSR, heat shock activates the Cell Wall Integrity (CWI) pathway, which is crucial for maintaining the structure and function of the cell wall under various stress conditions[15]. This pathway is primarily activated by environmental stresses such as heat shock, osmotic stress, and mechanical damage. At the core of the CWI pathway are membrane-bound mechanosensors like Wsc1 and Mid2, which detect cell wall stress and initiate a signaling cascade. These sensors activate Protein Kinase C (Pkc1), which then triggers a MAP kinase cascade involving Bck1, Mkk1/2, and Mpk1. This cascade leads to the activation of transcription factors such as Rlm1 and Swi4/Swi6, which promote gene expression in cell wall synthesis and remodeling[16]. Although activation of Mpk1 occurs relatively fast (approximately 15 min), Pkc1 pathway-mediated transcription occurs over much longer time periods, with genes such as *PRM5* and *FKS2* being maximally induced 8 h after the initial stress[17,18]. In contrast to metazoans, yeast only has one PKC isoform, Pkc1, which is essential for cell survival. This essential nature derives from the inability of *pkc1Δ* cells to maintain their cell wall structure and osmotic balance. Correspondingly, *pkc1Δ* cells can be kept viable on media supplemented with osmotic stabilizers such as sorbitol[18]. Although a large number of Pkc1-mediated effects occur through MAP kinase components such as Mpk1, it is now clear that[19] Pkc1 regulates numerous other targets[20]. Although different PKC isoforms display slightly varied substrate sequence specificity, it is commonly accepted that the general motif comprises basic amino acids (Arg or Lys) at positions −2 and +2 relative to the phosphorylated residue[20–22].

While the HSR and CWI pathways have been long established as key for the response to heat, more recently, the importance of protective protein phase separation has been highlighted. Heat causes protein misfolding and aggregation and, in eukaryotic cells, triggers the aggregation of proteins and RNA into biomolecular condensates[23]. Biomolecular condensates are adaptive, membrane-less structures of concentrated biomolecules that differ from protein-misfolded aggregates. These include stress granules, which contain ribosome-associated mRNAs and translation machinery components, and processing bodies (P-bodies), which accumulate translationally inactive mRNAs lacking initiation factors[24–27]. P-bodies and stress granules (SGs) are distinct organelles that are thought to be functionally linked. Recent evidence suggests that pre-existing P-bodies may serve as nucleation sites for SG assembly[28]. SGs maintain a dynamic equilibrium with polysomes and show sensitivity to translation inhibitors, functioning as crucial triage centers that direct mRNAs toward translation, storage, or degradation[29]. Poly-A-binding protein (Pab1) is a defining marker of stress granules[30]. Stress granules are endogenous protein aggregates that are disassembled without degradation during recovery. In addition to this relatively slow transcriptional response (30+ mins), recent studies have demonstrated the rapid formation of reversible quinary protein aggregates in response to acute heat exposure that occurs after only a few minutes[31]. The second type of biomolecular condensate-P-bodies specifically function as concentration hubs for translationally inactive mRNPs, housing machinery for both translation suppression and RNA degradation, and contain a fundamental set of conserved proteins[26]. Currently, the molecular mechanisms for this rapid response and the effect of heat shock on Hsp70 post-translational modification remain unknown.

A final universal paradoxical feature of the heat shock response is that while heat induces the expression of heat shock proteins, it overall leads to a global cessation of translation. This translational arrest is a protective mechanism that conserves energy and resources while the cell deals with the immediate threat posed by elevated temperatures. During heat shock, the dissociation of Hsp70 from ribosomal components plays a pivotal role in halting translation. This dissociation causes a pause in translational elongation, preventing the accumulation of misfolded proteins that could overwhelm the chaperone machinery[32]. In addition, specific kinases, such as Gcn2, are activated by stress and phosphorylate the alpha subunit of eukaryotic initiation factor 2 (eIF2α), further inhibiting translation initiation[33]. Heat shock has also been recognized to impair protein synthesis, affecting the elongation phase[34–36].

As proteomic technologies have improved, a large number of post-translational modifications (PTMs) have been discovered on molecular chaperones, collectively known as the Chaperone Code. This code fine-tunes chaperone functions by altering their interactions with client and co-chaperone proteins. Although a number of Hsp70 PTMs have been described, less than ten have been fully defined in terms of regulation and physiological relevance[1]. In this study, we seek to understand the interplay between the Hsp70 chaperone code and proteostasis. We identify a rapidly heat-induced phosphorylation site on yeast Hsp70 at T492 that Pkc1 mediates during heat shock. Phosphorylation of T492 promotes epichaperome remodeling, which drives the dissociation of Hsp70-Hsf1 and the activation of HSR. T492 phosphorylation also alters Hsp70-ribosome interactions, explaining the long-established observation of heat-induced translational pausing. Finally, we identify the CWI protein Bck1 as a bona fide client of yeast Hsp70 whose activity is dependent on T492 status. Together, our data provide a paradigm for how cells can coordinate activation of diverse signaling pathways that respond to heat through modulation of the Hsp70 chaperone code.

## Results

### Heat shock induces rapid phosphorylation of Ssa1 T492 in yeast

In an effort to understand how heat shock alters Ssa1 phosphorylation, yeast expressing FLAG-tagged Ssa1 were grown to the exponential phase and then treated for 1 h at 39 °C. Ssa1 was purified and subjected to mass spectrometry analysis. We identified a single site, threonine T492 (T492) that was induced upon heat shock. This residue is located in the substrate-binding domain of Ssa1 in close proximity to the region responsible for client binding (Fig. 1a). T492 is surprisingly well-conserved, indicative of this residue being important for chaperone function (Fig. 1b). To validate our mass spectrometry data, we generated a custom phospho-specific antibody for the T492 site and analyzed phosphorylation of T492 in yeast exposed to heat shock over a period of 4 h. To remove the confounding effects of the highly similar Ssa2, 3, and 4 proteins, we used the well-established MH272 *ssa1-4* strain, in which all four *SSA* genes have been deleted and functionally complemented by *SSA1* expressed from a *URA3*-marked centromeric plasmid[37]. The strain was transformed with a *LEU2*-marked plasmid bearing either wild-type FLAG-tagged Ssa1 or the non-phosphorylatable mutant ssa1-T492A. The *URA3* plasmid was evicted on 5-fluoro-orotic acid (5-FOA) media to yield strains expressing wild-type Ssa1, Ssa1-T492A as the sole Ssa protein in the cell, hereafter referred to as FLAG-WT or T492A cells. FLAG-WT or T492A cells were grown to mid-log phase and then exposed to 39 °C. Aliquots of cells were taken after 0, 1, 2 and 4 h, at which T492 phosphorylation was assessed via Western Blotting. Ssa1 T492 phosphorylation was minimal in unstressed cells, but was observed in cells heat-stressed for 1 h (Fig. 1c). Validating the specificity of the phosphoT492 antibody, no bands were seen in T492A lysates (Fig. 1c). To examine whether T492 phosphorylation occurs at shorter time scales, we examined T492 phosphorylation at sub-60 min time points. A robust increase in T492 phosphorylation was observed even after 1 min, suggesting this response represents an early stage of the heat shock response (Fig. 1d and Supplementary Fig. 1).

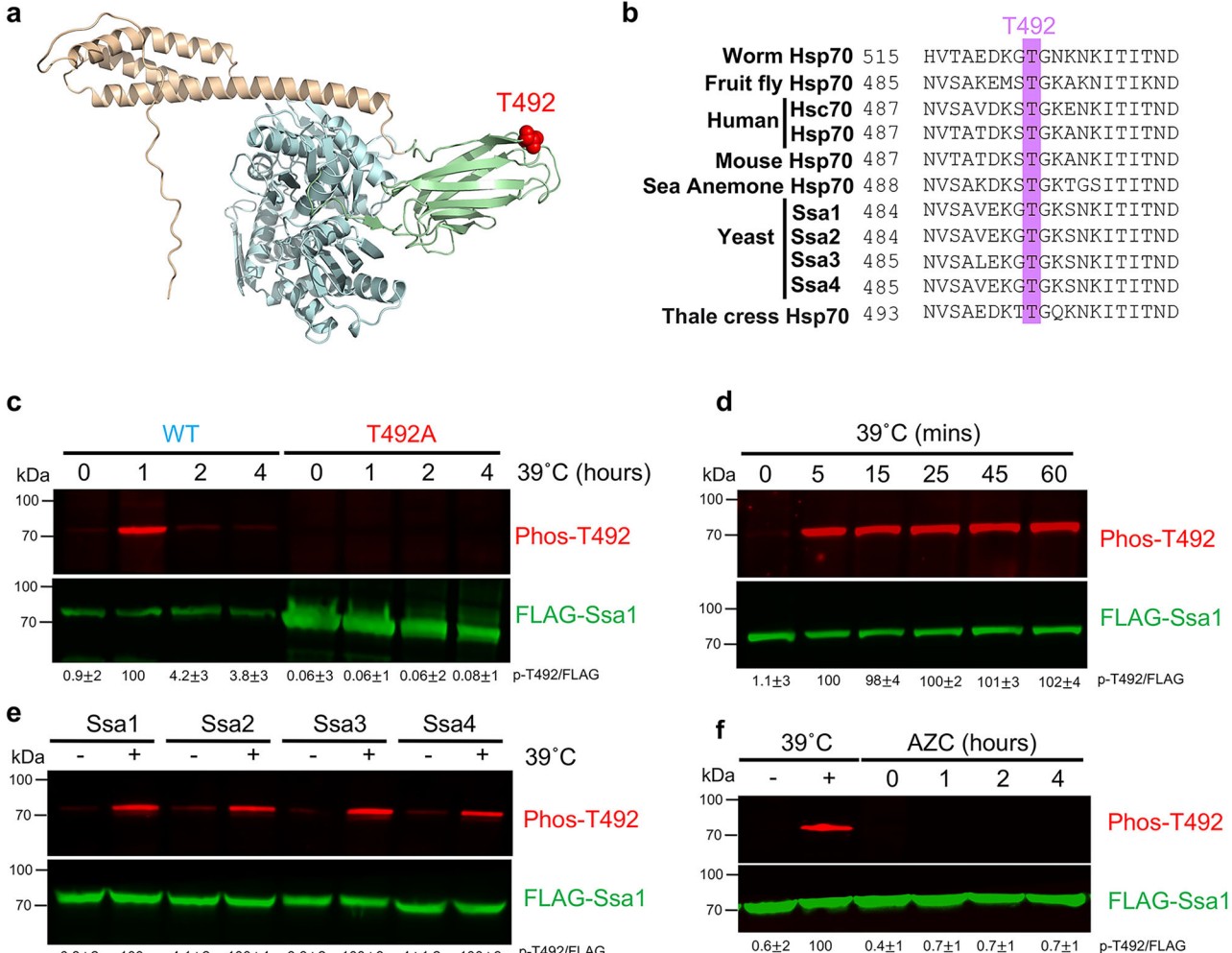

**Fig. 1 | Threonine 492 (T492) on yeast Hsp70 is activated in response to heat stress. a** T492 lies in the client binding domain of yeast Ssa1. T492 is highlighted on the closed predicted structure of Ssa1. **b** T492 is well conserved. Sequences of several Hsp70s are aligned, demonstrating the conservation of T492 in different eukaryotic species. **c** T492 is phosphorylated in response to heat stress. Lysate from WT or T492A cells treated at 39 °C for the indicated times were analyzed by Western Blotting using antisera to either FLAG or phospho-T492. **d** Phosphorylation of T492 is rapid in response to heat shock. Lysate from FLAG-WT cells treated at 39 °C for the indicated times were analyzed by Western Blotting using antisera to either FLAG or phospho-T492. **e** Ssa1, 2, 3 and 4 are phosphorylated in response to heat stress. Lysate from cells expressing either FLAG-Ssa1, Ssa2, Ssa3 and Ssa4 as the sole Ssa protein were treated at 39 °C for 1 h and t analyzed by Western Blotting using antisera to either FLAG or phospho-T492. **f** T492 phosphorylation is independent of protein misfolding. Lysate from FLAG-Ssa1 cells exposed to 39 °C or the proline homolog Azetidine-2-carboxylic acid (AZC) were analyzed by Western Blotting using antisera to either FLAG or phospho-T492. Each experiment involving western blotting was repeated three times with similar results. Source data are provided as a Source Data file.

## The Ssa isoforms are also phosphorylated in response to thermal stress

The four cytosolic Hsp70s in yeast Ssa1, 2, 3 and 4 are highly similar in amino acid sequence and appear to have at least some functional overlap[3,38]. Given that the T492 site is conserved in Ssa1-4, we queried whether T492 is phosphorylated in all Ssa1, 2, 3 and 4. We first generated yeast expressing either Ssa1, 2, 3, 4 on constitutive promoters as the sole Ssa in the cell and assessed T492 phosphorylation in these cells after 1 h of heat shock at 39˚ °C. Excitingly, heat-induced T492 phosphorylation was observed in Ssa1-4 (Fig. 1e). This identifies a conservation of a phosphorylation site between yeast Hsp70 paralogs.

## Heat-shock induced phosphorylation of T492 is independent of protein unfolding

Heat elicits a variety of effects on cells, including protein denaturation. To determine whether heat-induced phosphorylation of Ssa1 was promoted by protein misfolding, we utilized Azetidine-2-carboxylic acid (AZC), a proline analog that is incorporated in proteins during protein synthesis and triggers protein unfolding[39]. Treatment of yeast

with AZC did not promote Ssa1 T492 phosphorylation even over extended time periods (up to 4 h), suggesting that protein unfolding is not the primary stimulus for T492 activation (Fig. 1f).

## Heat-induced T492 phosphorylation is dependent on the Mid2 mechanosensor

Heat triggers activation of the yeast cell wall integrity MAPK pathway (Fig. 2a)[16]. To determine whether activation of Ssa1 phosphorylation was dependent on CWI signaling, we assessed whether T492 phosphorylation was impacted by loss of the Wsc1 or Mid2 mechanosensors[39,40]. Although heat-stimulated Ssa1 T492 phosphorylation was unchanged in a *wsc1Δ* strain, cells lacking Mid2 were unable to induce T492 phosphorylation (Fig. 2b). This result is consistent with those in Fig. 1f, as AZC does not activate Mid2 or the rest of the cell integrity pathway[40]. Heat activates the CWI by causing stretching of the plasma membrane[41,42]. To determine if heat-activated phosphorylation of Ssa1 T492 was caused through a similar mechanism, we treated yeast with the membrane stretching agent chlorpromazine (CPZ) and assessed T492 status (Fig. 2c). Robust T492 phosphorylation was

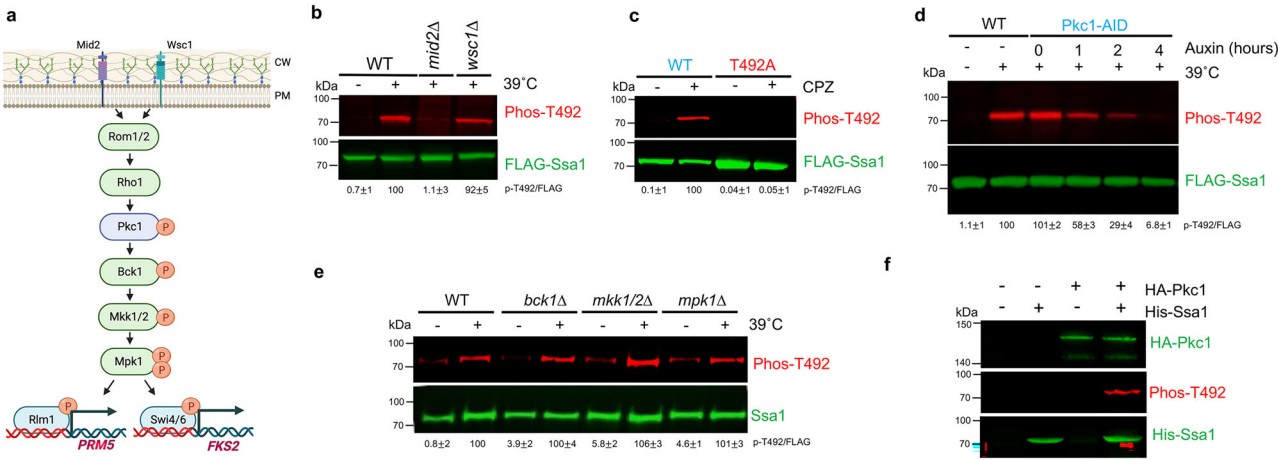

**Fig. 2 | Impact of cell integrity signaling on phosphorylation of T492. a** The yeast cell wall integrity pathway. Created in BioRender. Truman, A. (2025) https://BioRender.com/ibts1wx **b** Impact of Mid2 and Wsc1 mechanosensors on T492 phosphorylation. Lysates from indicated cells were analyzed by Western Blotting using antisera to either FLAG or phospho-T492. **c** T492 phosphorylation is triggered by membrane stretch. Lysate from FLAG-WT or T492A cells either untreated or treated with chlorpromazine (CPZ) were analyzed by Western Blotting using antisera to either FLAG or phospho-T492. **d** T492 phosphorylation is dependent on Pkc1. FLAG-Ssa1 (WT or with genomically AID-tagged Pkc1) cells were treated with auxin for the indicated times followed by 30-minute incubation at 39 °C. Lysates

were analyzed by Western Blotting using antisera to either FLAG or phospho-T492. **e** T492 phosphorylation is independent of Bck1, Mkk1, Mkk2 and Mpk1. Lysate from indicated knockout strains were analyzed by Western Blotting using antisera to either FLAG or phospho-T492. **f** Pkc1 directly phosphorylates Ssa1 T492. An in vitro kinase assay was performed with HA-Pkc1 purified from heat-shocked yeast and recombinant Ssa1. The reaction mixture was analyzed by Western Blotting using antisera to HA, phospho-T492 and HIS epitope. Each experiment involving western blotting was repeated three times with similar results. Source data are provided as a Source Data file.

observed after 1 hr of CPZ treatment (Fig. 2c). Taken together, these data suggest that T492 phosphorylation is not mediated by protein misfolding, but by membrane stretch, detected through the Mid2 mechanosensor.

## Yeast Protein Kinase C (Pkc1) directly phosphorylates Ssa1 T492

The requirement of Mid2 for T492 phosphorylation suggested the potential involvement of CWI kinases Pkc1, Bck1, Mkk1/2 and Mpk1 in phosphorylating Ssa1 (Fig. 2a). Several indirect lines of evidence implicated Pkc1 as the potential kinase for Ssa1, including the observation that overexpression of a hyperactive mutant of Pkc1 resulted in an increase in T492 phosphorylation[20,21,43]. In addition, Ssa1 has an arginine at the -3 position and a basic residue at the +2 position relative to the T492 site, which is the preferred substrate sequence for PKC proteins. To examine whether T492 phosphorylation was catalyzed by Pkc1, we decided to assess T492 phosphorylation when Pkc1 function was ablated. As the Pkc1 function is essential for cell viability, we utilized the auxin-inducible degron (AID) system[44]. After genomically tagging Pkc1 on the C-terminus with AID, we checked the system's functionality by plating the WT and Pkc1-AID cells on media containing or lacking auxin. Although both strains grew on YPD, only WT was able to grow on media containing auxin. The Pkc1-AID strain was, however, able to grow on auxin media when osmotic support was provided in the form of sorbitol (Supplementary Fig. 2a). These results were consistent with Pkc1 being degraded in the presence of auxin. We grew the Pkc1-AID strain to mid-log phase at 25 °C and then added auxin to ablate Pkc1 function. For each indicated time point, yeast were shifted to 39 °C for 30 min, and then T492 status was assessed via Western Blotting. Robust T492 phosphorylation was observed post-heat shock in WT and Pkc1-AID pre-auxin addition (Fig. 2d). Post-auxin addition, the ability of heat shock to stimulate T492 phosphorylation diminished over time, being barely observable 4 hrs after auxin treatment (Fig. 2d). This loss of T492 phosphorylation corresponded with a decrease of Pkc1-AID abundance (Supplementary Fig. 2b). Pkc1 is part of a kinase cascade consisting of Pkc1, Bck1, Mkk1/2 and Mpk1 (Fig. 2a). To rule out the possibility that Pkc1-dependence of T492 phosphorylation was via these other kinases, we assessed T492 phosphorylation in cells lacking

either Bck1, Mkk1/2 or Mpk1 (Fig. 2e). Cells lacking these kinases were still able to activate T492 phosphorylation in response to heat stress, indicating that Pkc1 is directly phosphorylating this site. To complement these results, we performed an in vitro kinase assay with purified Pkc1 and recombinant Ssa1. Purified Pkc1 was able to phosphorylate Ssa1 on T492 (Fig. 2f). Taken together, these data suggest that Pkc1 is directly able to phosphorylate Ssa1 on T492 in response to heat shock.

## Phosphorylation of T492 fine-tunes the Hsp70 epichaperome

Interactions of chaperones (epichaperome) are highly dynamic and vary depending on internal and external cues all of which modulate the chaperone code[1,45–50]. We considered the possibility that phosphorylation of Ssa1 at T492 may dictate the breadth and specificity of the Ssa1 epichaperome. We purified the Ssa1 complexes from heat-shocked WT and T492A cells and then compared their composition using mass spectrometry (Fig. 3a). The interacting proteins that are significantly downregulated were defined as $p < 0.05$ and have a log2 fold change $< -1$. For the significantly upregulated proteins, they have $p < 0.05$ and a log2 fold change $> 1$. We identified 647 proteins (1% FDR) and quantified 380 of those, which were significant. Of the total interactors identified, 24% displayed a preference for binding to WT Ssa1, 23% displayed a preference for binding to T492A samples, and 54% showed an equal affinity for both versions of Ssa1 (Fig. 3b). To determine the nature of the protein interactions altered by T492 phosphorylation we performed Gene ontology (GO) analysis. GO analysis revealed significant enrichment of multiple cellular functions, several of which were unique to either WT or T492A samples (Fig. 3c). The WT Ssa1 interactome displayed a preference for proteins involved in ribosome translation, protein trafficking and cytoskeletal organization (Fig. 3c). In contrast, the T492A mutant showed a preference for proteins involved in proteostasis and metabolism (Fig. 3c). Protein interactions can be dictated by protein localization. GO analysis of interactor localization revealed subtle changes between WT and T492A samples. WT Ssa1 displayed a binding preference for proteins in the endomembrane system and cell cortex. In contrast, the T492A mutant selectively preferred proteins found in the extracellular region

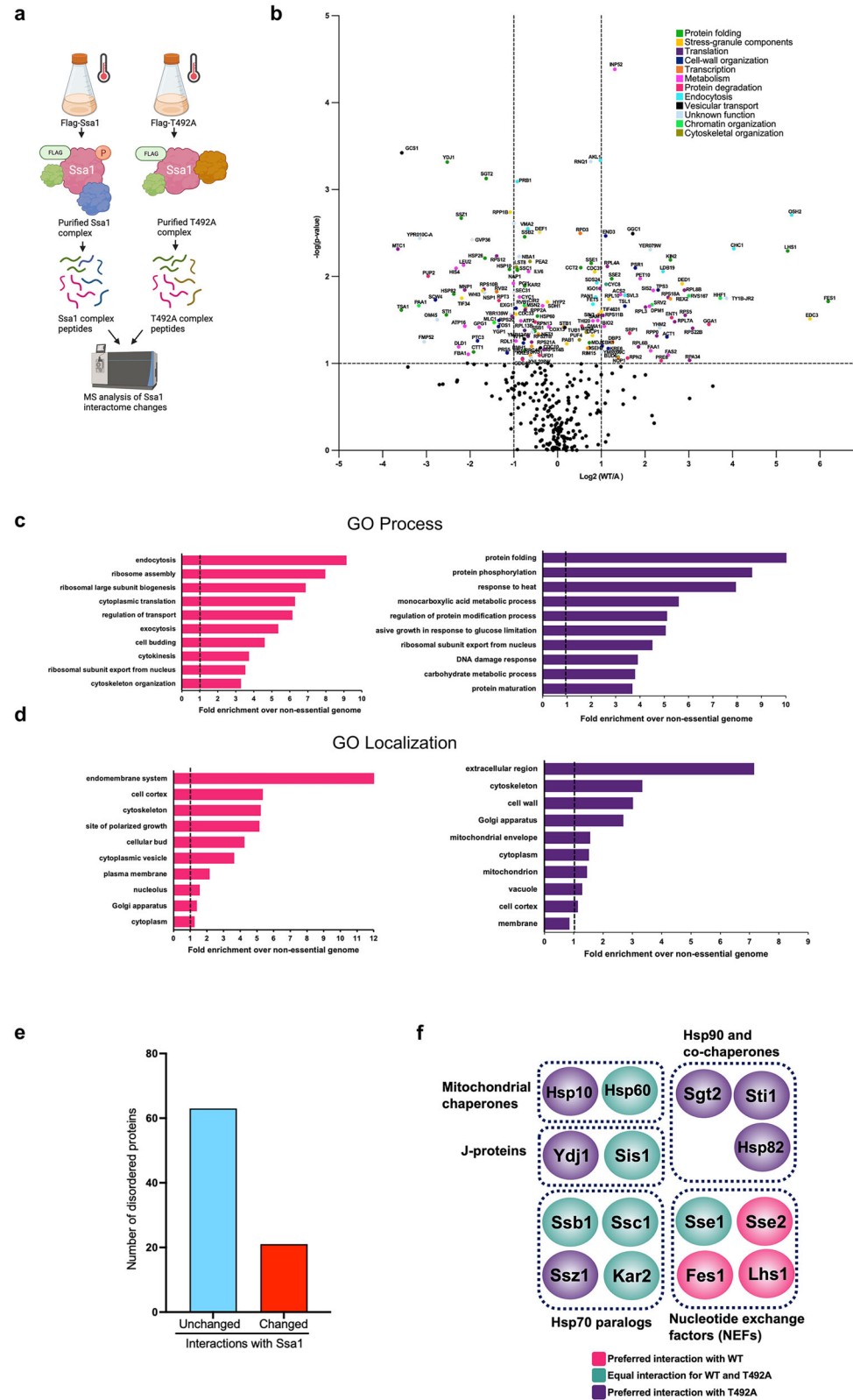

and cell wall (Fig. 3d). Both yeast Hsp70 and Hsp90 bind intrinsically-disordered regions (IDRs) on proteins[47,51]. To understand whether T492 phosphorylation may impact the ability of Ssa1 to bind IDRs, we batch analyzed clients for the percentage of intrinsically disordered amino acids via MobiDB. There were roughly threefold fewer disordered proteins in T492-dependent interactors (Fig. 3e). The activity and specificity of Ssa1 is impacted through interaction with co-

chaperone proteins that include Sis1 and Ydj1[52–58]. We considered the possibility that the altered clientome of Ssa1 in T492A may also be an effect of differential co-chaperone/chaperone binding. In our proteomics study, mutation of T492 impacted the association of nine key chaperone and co-chaperone proteins (Fig. 3f). These included the major Hsp40-Ydj1, nucleotide exchange factors-Sse2, Fes1, and Lhs1. Interestingly, the T492A mutant displayed enhanced binding with the

**Fig. 3 | Ssa1 T492 phosphorylation fine-tunes the epichaperome. a** Proteomic workflow. WT and T492A Ssa1 complexes were purified from heat-shocked yeast, and epichaperome changes were quantified using mass spectrometry. Created in BioRender. Truman, A. (2025) https://BioRender.com/24ppob4 **b** Volcano plot of epichaperome changes between WT and T492A cells. The log2 change (WT/T492A) and p-value for each interactor are plotted from the values obtained from 3 biological replicates. Statistical significance was determined by ANOVA. **c** GO analysis of WT vs T492A epichaperome based on cellular processes. **d** GO analysis of WT vs

T492A epichaperome based on cellular localization. **e** T492 phosphorylation preferentially alters interaction with proteins of low disorder. Number of disordered proteins that significantly changed interaction with Ssa1 (log2 WT/A value > 1 or < −1) vs unchanged. **f** T492 phosphorylation alters the interaction between Ssa1 and major chaperone and co-chaperone proteins. The 15 chaperones and co-chaperones were grouped by homology and function and then colored based on the interaction change between WT and T492A samples.

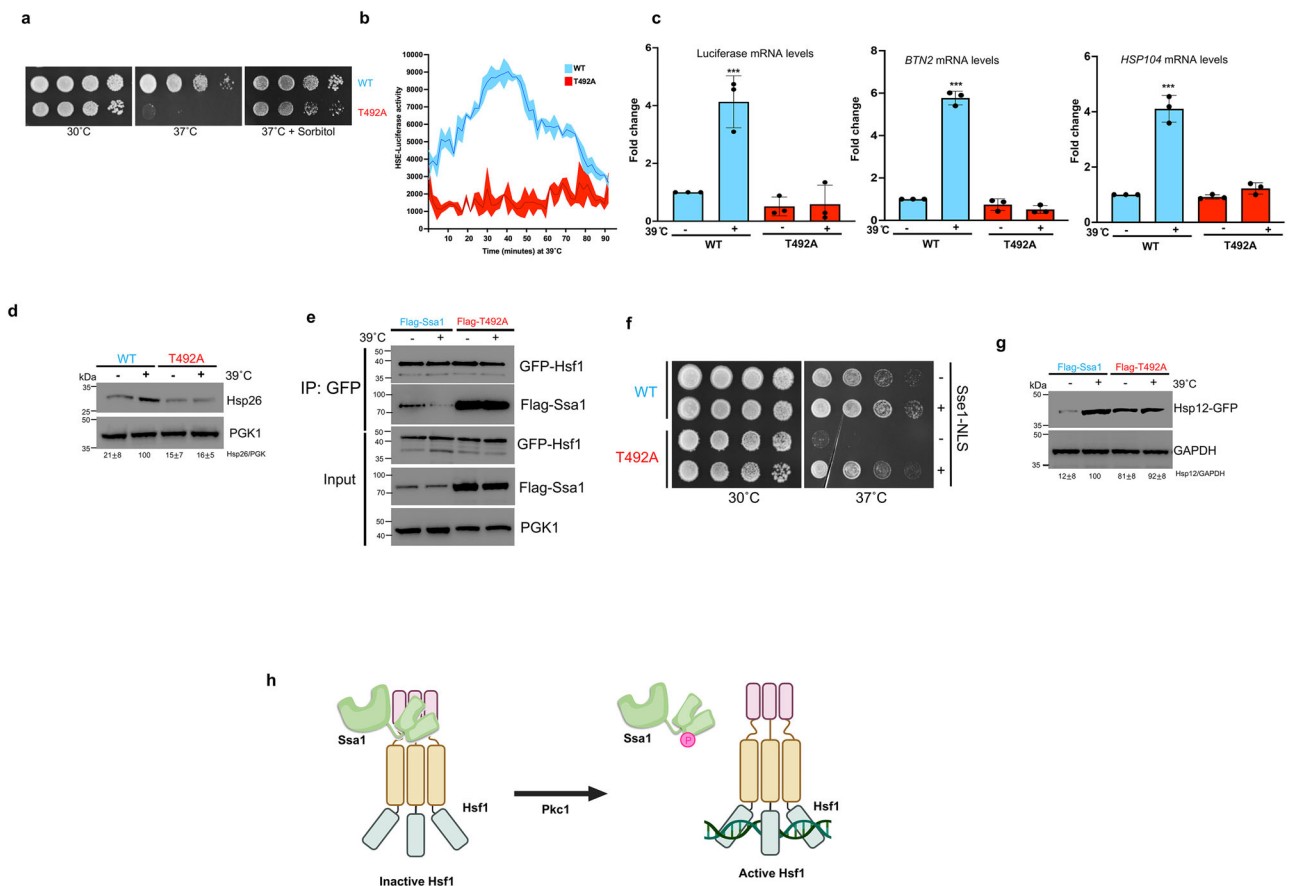

**Fig. 4 | T492 phosphorylation controls the heat shock response in yeast.**
**a** T492A cells are temperature-sensitive, a phenotype that can be suppressed with osmotic stabilization. WT and T492A cells were tenfold serially-diluted onto the indicated media and photographed after 3 days. **b** HSE-luciferase activity of WT and T492A cells. Cells were transformed with a real-time destabilized secreted HSE-luciferase reporter. Cells were incubated at 39 ˚C in a Synergy H1 reader for the indicated times, and luciferase activity was measured. Values shown represent the mean, minimum and maximum luciferase activity normalized to cell number for five replicates. **c** Luciferase, *HSP104* and *BTN2* transcript expression in WT and T492A cells. We measured the transcription of the luciferase gene, *HSP104* and *BTN2* in WT and T492A cells under unstressed and heat-shocked conditions via RT-PCR. Gene expression was normalized against that of *ACT1* in each strain, and the resulting ratios in WT cells were defined as onefold. Data are presented as mean and standard deviation 3 biological replicates. Statistical significance was determined by two-way ANOVA with a multiple comparisons test. ***$p \leq 0.001$. **d** Heat-induced Hsp26 induction is impaired in T492A cells. Lysate from FLAG-WT or T492A cells

treated at 39 °C for 30 min were analyzed by Western Blotting using antisera to either Hsp26 or PGK1. **e** T492 phosphorylation antagonizes the Ssa1-Hsf1 interaction. GFP-Hsf1 was purified from FLAG-WT or T492A yeast under either untreated or heat-shocked conditions using GFP-Trap beads. Lysates and immunoprecipitates were analyzed by Western Blot using antisera to GFP or FLAG epitopes and PGK1 as a loading control. **f** Overexpression of nuclear Sse1 suppresses the *ts* phenotype of T492A cells. WT or T492A cells transformed with plasmids for Sse1 or Sse1-NLS were tenfold serially diluted onto the indicated media. Plates were photographed after 3 days. **g** T492 phosphorylation impacts Msn2/4-mediated transcription. Lysates from FLAG-WT and T492A cells expressing genomically-tagged Hsp12 were assessed by Western Blot using antisera to GFP or GAPDH. Each experiment involving western blotting was repeated three times with similar results. Source data are provided as a Source Data file. **h** Summary figure. Heat-mediated T492 phosphorylation of Ssa1 drives Ssa1-Hsf1 dissociation. Created in BioRender. Truman, A.(2025) https://BioRender.com/usi7p63.

ribosomal-associated-Ssz1, the Hsp90 paralog Hsp82, and its co-chaperones Sgt2 and Sti1 (Fig. 3f). Despite this major remodeling of the Ssa1 interactome upon mutation of T492, Hsp40-Sis1, other Hsp70 paralogs-Ssb1, Ssc1, Kar2, and the mitochondrial chaperone Hsp60 showed equal interaction with WT and mutant under heat shock (Fig. 3f).

## Phosphorylation of Ssa1 T492 is critical for correct activation of the Heat Shock Response

Molecular chaperones are critical for the refolding of heat-denatured proteins and thus survival at high temperatures. Although viable at 30 ˚C, T492A cells were unable to grow on solid media at 37 ˚C, suggesting a possible defect in the heat shock response (Fig. 4a). To

investigate this possibility, we assessed the expression of an HSE-driven destabilized luciferase reporter[11] in WT and T492A cells. WT cells exposed to 39 °C mounted a rapid and robust HSR within minutes, reaching a maximum after 30 minutes of heat exposure. In contrast, T492A cells did not induce significant expression of the HSE-luciferase construct (Fig. 4b). To confirm that the loss of HSE-reporter activity in T492A was due to transcriptional rather than translational effects, we measured the transcription of the luciferase gene in these cells under unstressed and heat-shocked conditions via RT-PCR. As expected, HSE-mediated luciferase transcription was compromised in the T492A mutant strain (Fig. 4c). We also determined the expression of *HSP104* and *BTN2*, genes whose transcription are highly-dependent on Hsf1 activity[59]. In agreement with our previous data, the heat-induced transcription of these genes were abolished in T492A cells (Fig. 4c). To complement our transcriptional data, we assessed induction of the well-characterized heat-inducible protein Hsp26 in WT and T492 cells. Hsp26 levels did not respond to exposure to 39 °C in T492A cells, confirming that T492 phosphorylation is critical for the heat shock response (Fig. 4d).

## T492 phosphorylation controls the interaction between Ssa1 and Hsf1

The lack of heat-induced transcriptional response in T492A cells suggested a defect in Hsf1 regulation. Ssa1 regulates the response to heat shock through its interaction with Heat Shock Factor, Hsf1[9–11,60–64]. To understand whether Ssa1-Hsf1 interaction was impacted by the T492A mutation, we measured Ssa1-Hsf1 affinity in WT and T492A cells under unstressed and heat shock-treated conditions via co-immunoprecipitation. As expected, in WT cells the interaction between Ssa1 and Hsf1 was disrupted by heat shock (Fig. 4e). In contrast, the Ssa1-Hsf1 interaction remained unchanged in T492A heat-shocked cells, providing a rationale for the defective HSR observed (Fig. 4e). Together with our proteomics showing unchanged Sis1 binding to Ssa1-T492A, these data indicate that Sis1-dependent assembly of the Hsf1–Hsp70 complex is intact, whereas Pkc1-driven Ssa1-T492 phosphorylation is required for rapid heat-induced release of Hsf1.

## Accelerating Hsp70 substrate release suppresses the temperature sensitivity of T492 cells

Release of Hsp70 from Hsf1 can be accelerated by overexpression of the nuclear-targeted Sse1 co-chaperone protein[12]. Our data suggested the primary cause of heat-sensitivity of T492 cells may be an inability for Ssa1 to release Hsf1. We expressed either WT Sse1 or Sse1-NLS in WT or T492 yeast and examined the ability of transformants to grow at elevated temperatures (Fig. 4f). In agreement with our hypothesis, the temperature-sensitivity of T492 cells was suppressed by overexpression of Sse1-NLS but not WT Sse1 (Fig. 4f).

## Elevated Ssa1 abundance in T492A cells is driven by altered transcriptional programs

The data from 4a-f clearly suggested a defect in Hsf1-driven transcription in the T492A mutant, potentially leading to decreased Ssa protein abundance. Paradoxically, throughout our experiments, we noticed that the steady-state level of the T492A mutant was substantially *higher* than that of WT (Figs. 1c, 2c and 4e). To determine whether the increased abundance of the T492A mutant could be attributed to transcriptional effects, we created yeast strains expressing FLAG-Ssa1 and FLAG-T492A from the constitutive GPD promoter. In these strains, FLAG-Ssa1 and FLAG-T492A were expressed at equal levels (Supplementary Fig. 3a). To ensure the abundance of T492A in Fig. 4a was not the primary driver of the *ts*-sensitive phenotype seen in Fig. 4a, we repeated the serial dilutions using our newly generated GPD-Flag-Ssa1 and GPD-Flag-T492A strains. The GPD-Flag-T492A was as sensitive to 37 °C as the SSA promoter equivalent, confirming that

the phenotype observed was promoter-independent (Supplementary Fig. 3b). To confirm that transcription of T492A was elevated in our native promoter strains, we performed RT-PCR. In accordance with our results, T492A transcription was elevated compared to WT in native promoter strains, but not in strains expressing FLAG-Ssa1 from the GPD promoter (Supplementary Fig. 3c and d). While physiologically relevant, we considered the possibility that the reinforced interaction between Hsf1 and Ssa1 in T492A may be driven by increased Ssa1 abundance rather than enhanced Ssa1-Hsf1 binding affinity. We revisited the IP experiment in Fig. 4e using the newly generated GPD-FLAG-Ssa1 strains. Although the abundance of WT Ssa1 and T492A was equalized in these strains, Hsf1 displayed a substantially enhanced interaction with T492A compared to WT (Supplementary Fig. 3e). Previous studies have shown that a subset of heat-induced genes are controlled by Msn2/4 and that inhibiting Hsf1 activity can induce Msn2/4-mediated transcription[65]. To query whether T492 status may also impact Msn2/4 activity, we examined the levels of Hsp12 (a reporter for Msn2/4-mediated transcription) in WT and T492A cells (Fig. 4g). T492A cells displayed a constitutive activity of Msn2/4 compared to WT, potentially explaining the increased transcription of SSA1 in the SSA-driven T492A strain (Fig. 4g). Taken together, these data demonstrate that T492 phosphorylation is important for Ssa1-Hsf1 dissociation upon heat shock and appropriate activation of heat shock-induced transcription (Fig. 4h).

## T492 phosphorylation fine-tunes Ssa1 ribosome engagement and translation fidelity during heat shock

Several of the Ssa1 interactors altered by T492 phosphorylation are important for protein translation, ribosome assembly, and translation initiation (Fig. 5a and b). Heat shock remodels Hsp70-ribosome interactions, and both transient dissociation and persistent over-association of Hsp70 have been reported to impede elongation, thereby contributing to global translational pausing[32]. Phosphorylation of Hsp70 on T495 (equivalent to T492 in yeast) by the *Legionella* Legk4 kinase during infection can trigger a similar effect to heat shock[66]. To determine whether the T492A mutation in yeast may alter translation, we quantified the extent of global translational repression during temperature stress in WT and T492A yeast using the ratio of polysome to monosome signal in polysome profiles in a manner similar to ref. 14. Both strains showed a similar decrease in translation upon a 10-minute stress at 39 °C, and both strains recovered to nearly unstressed levels after 30 minutes at 25 °C (see Fig. 5c). In order to track chaperone association with translating ribosomes during stress, we purified proteins from the polysome fractions and quantified the abundance of chaperones by western blotting. While at 25 °C, chaperone association with polysomes was similar across strains, after 10 minutes at 39 °C, levels of polysome-bound Ssa1-T492A increased while levels of polysome-bound WT Ssa1 and other chaperones remained steady (Fig. 5d). In order to quantify overall protein synthesis, we measured incorporation of the methionine analog L-homopropargylglycine (HPG) into newly synthesized proteins over the course of 30 minutes at 25 °C or 39 °C (see Fig. 5e)[68]. Control cells were additionally incubated with cycloheximide to prevent translation. At the specified time points, cells were fixed with ethanol, and then the HPG was modified with a fluorescent dye using a copper-chelating click reaction[67]. The fluorescence intensity of each cell was then measured by flow cytometry. At both temperatures, the Ssa1-T942A strain had lower translation rates than WT as measured by HPG incorporation ($p = 0.0005$ by an analysis of covariance test). If T492 phosphorylation is required to fine-tune Ssa1-ribosomal protein interactions during heat stress, we considered the possibility that the T492A mutant may disrupt translational fidelity. We assessed translational fidelity in WT and T492A mutant cells using a well-established dual luciferase assay. In this system, *Renilla* luciferase (Rluc) and firefly luciferase (Fluc) are under the control of separate constitutive promoters[51,68]. The Fluc

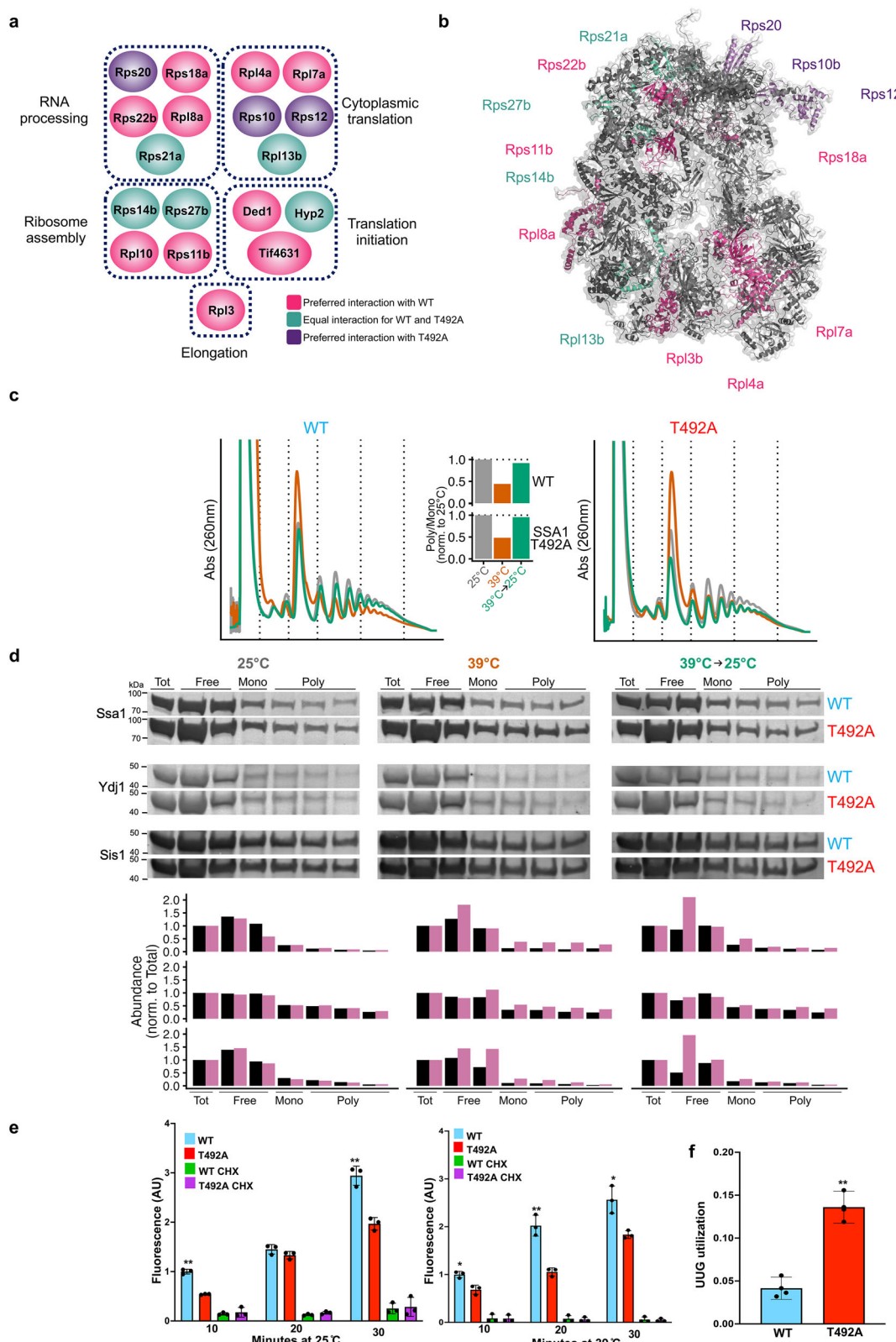

mRNA has a non-traditional start codon (UUG), which is only used when there is a loss of translational fidelity[51]. The Rluc mRNA has a classic AUG start codon and therefore acts as a control for alterations in overall translation. Using this system, we observed an approximately threefold increase in utilization of the non-canonical start codon in T492A cells (Fig. 5f). Taken together, these data suggest that loss of T492 phosphorylation causes persistent Ssa1 association with

elongating ribosomes and a measurable drop in codon-specific translation fidelity.

### T492 phosphorylation is involved in P-body disassembly

Our epichaperome analysis revealed several protein interactors that comprise stress granules, P-bodies or both (Figs. 3 and 6a). Both of these types of cytoplasmic RNA-protein granules are induced under

**Fig. 5 | T492 phosphorylation impacts heat-induced Ssa1-ribosome association and translational fidelity. a** mutation of T492 alters the interaction of Ssa1 with translational machinery. Data shown are taken from Fig. 3b. **b** ribosomal subunits displaying altered interaction upon T492 mutation. Structure based on PDB file 4V7R. **c** Polysome profiles of WT and T492A yeast under untreated and heat-shocked and recovery conditions. **d** Interaction of Ssa1, Ydj1 and Sis1 with polysome fractions in untreated, heat-shocked and recovery conditions. Each experiment involving western blotting was repeated three times with similar results **e** Translation was measured by incubating cells with the methionine analog homopropargylglycine (HPG) for the indicated time and then fluorescently labeling the cells with click chemistry. Control cells were incubated with the translation inhibitor cycloheximide (CHX). Statistical analysis was performed using two-way ANOVA. *$p \leq 0.05$, **$p \leq 0.01$. Data plotted is the mean fluorescence intensity of the population, normalized to WT (10 min) for each temperature. **f** Translational fidelity of WT and T492 cells was determined using a UUG/AUG dual luciferase assay as described in ref. 51. The data shown are the mean and standard deviation of three biological replicates. Statistical significance was calculated via a two-sided unpaired $t$ test. **, $P \leq 0.01$. Source data are provided as a Source Data file.

heat stress[23,31,69–71]. We queried whether the T492 phosphorylation state may impact stress granule and P-body formation by tracking the localization of Edc3-mCherry (P-body marker) and Pab1-GFP in WT and T492A cells. Despite previous observations that the resolution of Pab1 requires Ssa1 and associated co-chaperones, the formation and resolution of Pab1 foci were not dependent on T492 phosphorylation (Fig. 6a–e). In contrast, while Edc3 foci formation was unaffected by T492 status, the kinetics of Edc3 foci disassembly were impaired in the T492A mutant during 60–120 min recovery (Fig. 6a–e).

### Heat-induced phosphorylation of yeast Hsp70 promotes CWI signal amplification

To understand the physiological impact of T492 phosphorylation, we screened WT and T492A cells for their resistance to a range of cellular stresses (Fig. 7a). While T492A cells grew at nearly WT rates at 30°C, they were nevertheless compromised for their resistance to temperature, caffeine, calcofluor white (CFW) and hydroxyurea (HU). These stresses are well-known activators of the Cell Wall Integrity (CWI) pathway, and consequently, we considered the possibility that T492 phosphorylation may also govern cell integrity signaling. We assessed activity of the CWI pathway in WT and T492 cells via *PRM5* and *FKS2* promoter-lacZ reporter assays as in refs. 17,73. T492A cells were unable to recapitulate WT levels of heat-induced *FKS2* and *PRM5* transcription (Fig. 6b, c). While both *PRM5* and *FKS2* expression are dependent on cell integrity, their modes of activation differ. *PRM5* expression is dependent on Rlm1 phosphorylation, catalyzed by the Mpk1 kinase[17,73]. In contrast, *FKS2* expression requires dual phosphorylation of Mpk1, but is independent of Mpk1 catalytic activity[72,74]. Consistent with our promoter-reporter results, T492A cells were defective in Mpk1 dual phosphorylation, but not Mpk1 stability (Fig. 7d). The upstream semi-redundant kinases Mkk1 and Mkk2 were not affected by the mutation of T492, so we focused our attention further upstream on the Bck1 kinase (Fig. 7e, f, respectively). To determine whether the point of failure in the cell integrity of T492A cells was at Bck1, we assessed Bck1-HA levels (driven by a constitutive promoter) in WT and T492A cells (Fig. 4g). Strikingly, the abundance of Bck1 was substantially lowered in T492A cells (Fig. 4g). Bck1 is a 147 kDa protein that consists of a kinase domain and a large intrinsically disordered domain of unknown function[75,76].

To determine if T492 phosphorylation preferentially impacted a particular domain of Bck1, we separately expressed the N- and C-terminal domains of Bck1 (amino acids 1–738 and 739–1478) each with an HA tag in WT and T492A cells (Fig. 7g). Although the abundance of Bck1 (1–738) was not impacted by T492 status, we observed a decrease in the steady-state levels of Bck1 (739–1478) in T492A cells upon heat shock (Fig. 7g). To determine whether Bck1 may be a client of Ssa1, we performed directed co-immunoprecipitation experiments of FLAG-Ssa1 against Bck1-HA (both fragments) in WT and T492A yeast. We observed a robust interaction of Ssa1 with both Bck1 fragments, independent of heat shock (Fig. 7h, i). Overall, these data identify a requirement for Ssa1 T492 phosphorylation in promoting CWI signaling through support of Bck1.

## Discussion

The core components of the heat shock response are well conserved throughout life; the expression of molecular chaperones is induced in a heat shock factor-dependent manner to prevent protein aggregation and denaturation. How cells fundamentally detect heat and how they respond so fast to changes in temperature have been a source of debate. A favored model is the titration model whereby the accumulation of denatured proteins triggers the recruitment of Hsp70 away from HSF, allowing HSF to become active and induce heat shock genes[9–12,63]. However, this model does not account for the rapid kinetics of Hsf1 activation observed in response to heat shock, which occur faster than the time required for significant protein unfolding and chaperone sequestration. These discoveries point to a more nuanced and integrated model of HSR regulation that includes direct sensing of stress signals and intricate interactions between chaperones, co-chaperones, and Hsf1. Thus, while the chaperone titration model has provided valuable insights into HSR activation, it likely represents only a part of a more complex and dynamic regulatory system. In this study, we have uncovered a paradigm of rapid yeast Hsf1 activation that occurs through phosphorylation of Ssa1. Heat acts to cause membrane stretch, which is detected by the Mid2 mechanosensor. Activation of the cell integrity pathway leads to Pkc1-catalyzed phosphorylation of Ssa1 at T492, promoting a variety of cellular effects, including Ssa1-Hsf1 dissociation. Notably, T492 phosphorylation occurs rapidly within minutes and is not triggered by protein misfolding, suggesting it lies adjacent to the traditional Hsf1-Ssa1-unfolded protein titration mechanism, consistent with studies that demonstrated the interaction between Ssa1 and Hsf1 at the Ssa1-interacting region situated to the N-AD between residues 50 and 100. This result is consistent with the observation that interaction between Hsp70 and Hsf1 under heat shock occurs via the C-terminal substrate binding domain of Hsp70, the location of T492[9–11,77,78]. We do acknowledge that there remains a small heat-dependent loss of Hsf1-Ssa1 interaction in the T492A, consistent with previously identified mechanisms of HSR that include the titration model and HSF as a direct sensor for heat. Our findings reconcile Sis1's established role in restraining Hsf1 activity with the unchanged Ssa1–Sis1 interaction in T492A: Sis1 promotes basal Hsf1 repression, while heat-activated Pkc1 phosphorylation of Ssa1 at T492 licenses Hsf1 release. The suppression of T492A temperature sensitivity by nuclear Sse1 further supports a defect in the release/nucleotide-exchange step rather than in Sis1-mediated assembly. The Hsf1-mediated Heat Shock Response (HSR) and the Environmental Stress Response (ESR) controlled by Msn2/4 overlap in the sets of genes they induce under heat shock, and inhibition of one can lead to upregulation of the other[65]. In agreement with these observations, the T492A mutant displayed a compromised HSR while upregulating ESR. This result may explain the increased abundance of the T492A protein vs WT even when expressed from the same *SSA*-driven promoter.

The reciprocal interplay between the cell wall integrity pathway and T492 phosphorylation uncovered in this study explains their previously observed co-activation and regulation in response to heat stress. Previous studies have identified peripheral roles for chaperones

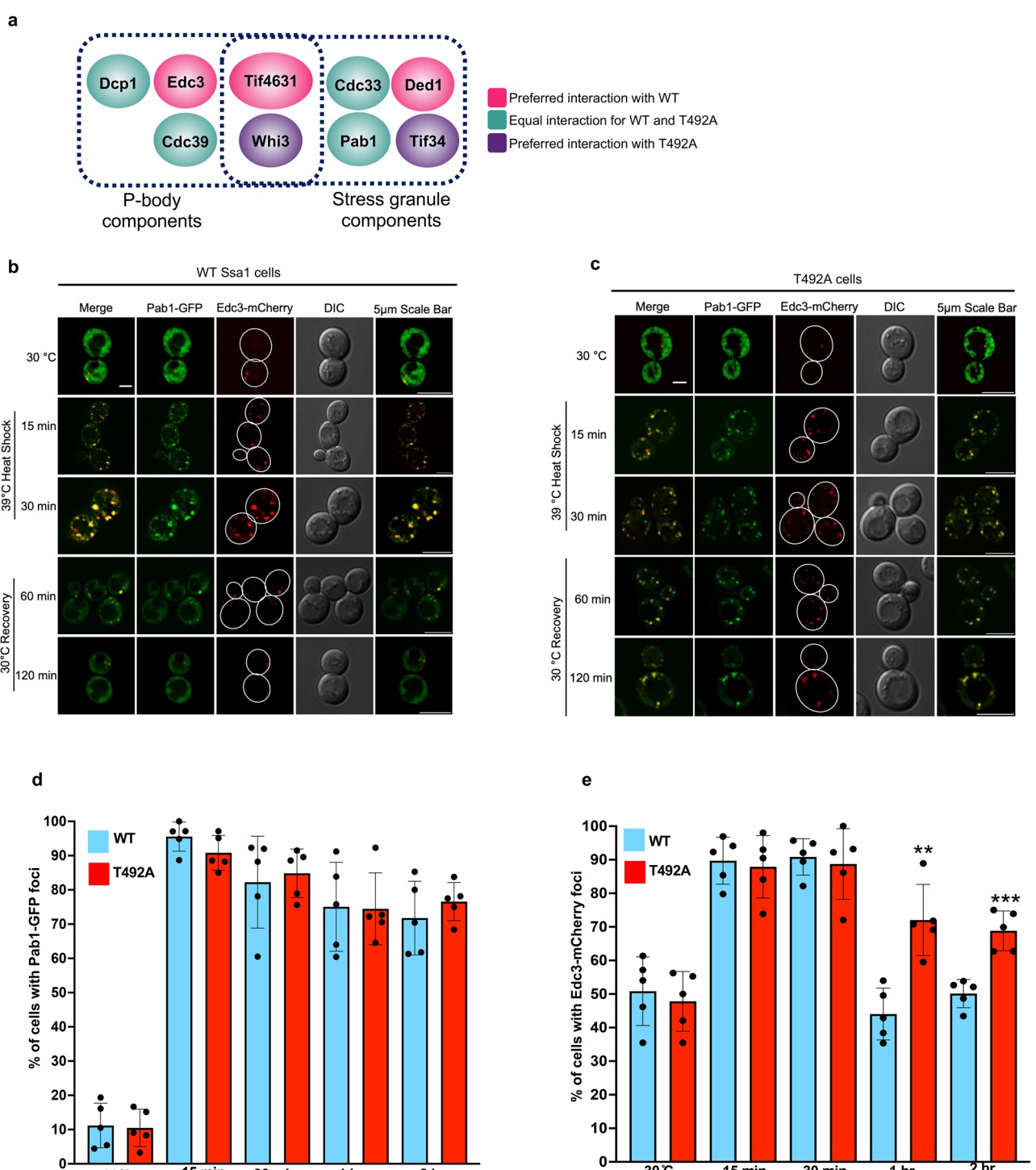

**Fig. 6 | T492 phosphorylation selectively alters P-body phase separation.**
**a** T492 phosphorylation alters interactions with key P-body and stress granule components. Data taken from 3b. **b** Edc3 and Pab1 foci formation and resolution in response to transient heat shock in WT, (**c**) Edc3 and Pab1 foci formation and resolution in response to transient heat shock in mutant. **d** T492 phosphorylation does not impact Pab1 foci resolution in response to heat shock. **e** T492 phosphorylation alters Edc3 foci resolution post-heat shock. The mean and standard deviation were calculated from three biological replicates ($n = 100$ cells per replicate). Scale bar indicates 5µm. Statistical significance determined via two-way ANOVA, **$p < 0.01$; ***$p < 0.001$. Source data are provided as a Source Data file.

in CWI. Hsp90 binds to the dually phosphorylated form of the Mpk1 MAP kinase, supporting its kinase activity, allowing it to phosphorylate Rlm1[48,79–82]. In addition, the slow growth phenotype of yeast lacking the yeast co-chaperone Ydj1 can be suppressed by overexpression of Mid2, the mechanosensor that we identify here as critical for activation of T492 under heat stress[83]. Our finding that Pkc1 directs a variety of

responses to heat shock through Ssa1 independent of the Bck1-Mkk1-Mpk1 MAPK cascade also may explain the differences in knockout phenotypes for these proteins. While loss of Pkc1 causes cell inviability, *bck1Δ* cells are viable but are temperature-sensitive[16]. A major challenge in understanding chaperone-kinase interactions is their reciprocal nature, as chaperones can stabilize the very kinases which then

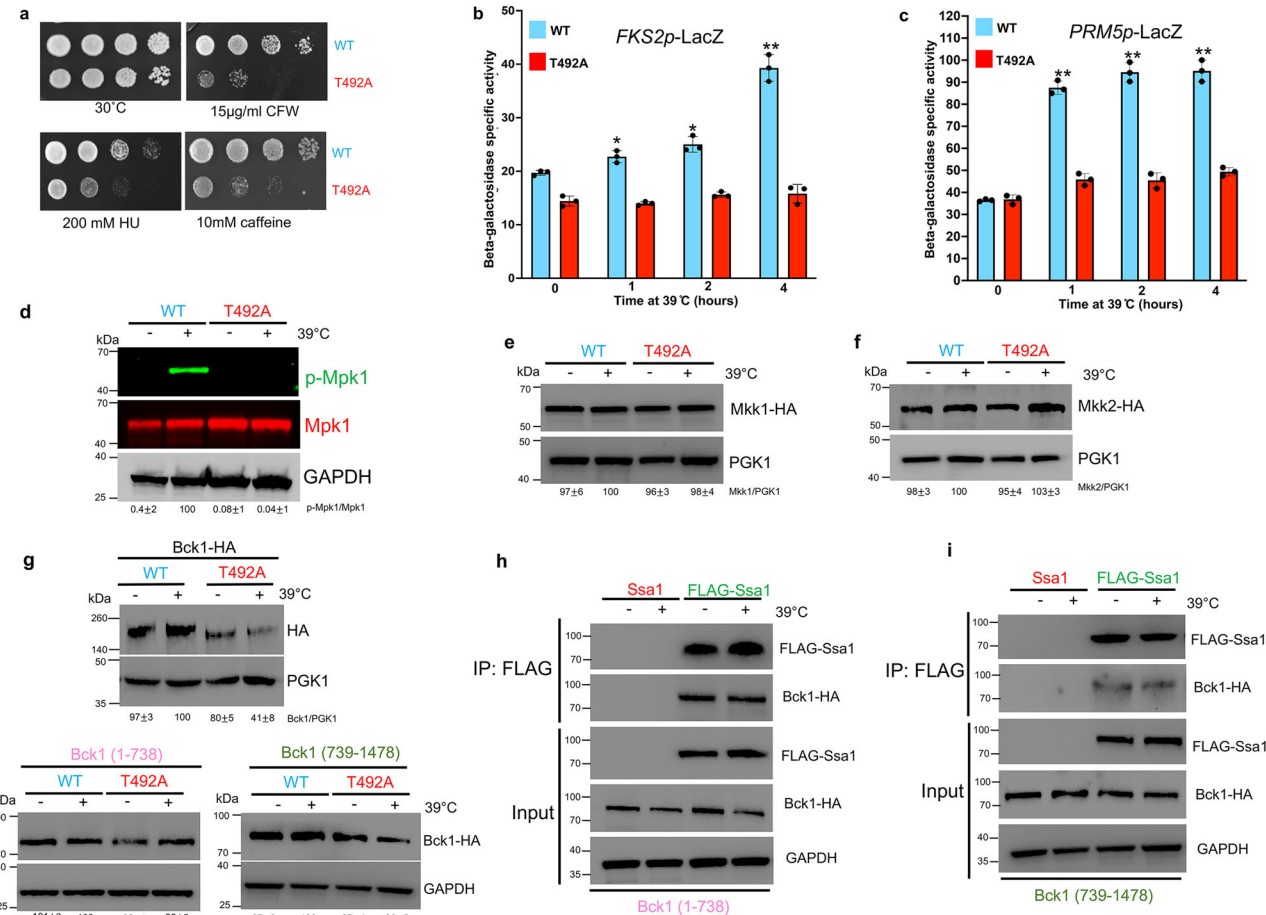

**Fig. 7 | T492 phosphorylation controls unique aspects of cell integrity signaling. a** T492A cells are sensitive to agents that perturb cell integrity. **b** *FKS2* expression is reduced in T492A cells. The data shown here are the mean and standard deviation of 3 biological replicates. Data are presented as mean ± standard deviation from $n = 3$ independent replicates Statistical significance determined via ANOVA, *$p < 0.05$; **$p < 0.01$. **c** *PRM5* expression is compromised in T492A cells. The data shown here are the mean and standard deviation of 3 biological replicates. Data are presented as mean ± standard deviation from $n = 3$ independent replicates

Statistical significance determined via ANOVA, *p < 0.05; **$p < 0.01$; **d** dual phosphorylation of the Mpk1 SAPK is abolished in T492Aa cells. **e** Stability of the Mkk1 MEK is independent of T492 phosphorylation status. **f** Stability of the Mkk2 MEK is independent of T492 phosphorylation status. **g** Stability of the Bck1 MAP3K is reliant on T492 phosphorylation. **h** The N-terminus of Bck1 interacts with Ssa1. **i** The C-terminus of Bck1 interacts with Ssa1. Each experiment involving western blotting was repeated three times with similar results. Source data are provided as a Source Data file.

phosphorylate and fine-tune chaperones[1,84]. In this study, we have demonstrated that while T492 phosphorylation of Ssa1 is dependent on upstream CWI components, T492 phosphorylation is also critical for the downstream activation of this same pathway. We identified the MAP3K Bck1 as an interactor and bona fide client of the yeast chaperone system. Bck1 is an unusual yeast MEKK in that while it contains a classic kinase domain, the rest of the 163 kDa protein has minimal structure and is likely intrinsically disordered. Our data underpin a role for T492 phosphorylation in promoting the stability and activity of the Bck1 C-terminal domain. Taken together our results underpin a non-canonical amplification of CWI signaling through Pkc1-mediated phosphorylation of Ssa1.

As newly synthesized yeast proteins exit the ribosome they are bound by ribosome-associated chaperones that include Ssb1, Ssz1 and Zuo1 to promote client folding[85–87]. However, the yeast ribosome itself is an extremely complex structure consisting of 4rRNA molecules as well as 79 ribosomal proteins[88]. Many of these proteins are highly prone to aggregation, and the process of assembling a ribosome must be tightly controlled to ensure translational fidelity. It is thus unsurprising that several studies have demonstrated chaperone control of the ribosome and translation. Mutation of the yeast Hsp70 co-chaperone Ydj1 leads to defects in translation of reporter proteins

and sensitivity to translation inhibitors such as hygromycin B and cycloheximide[89]. More recently, two crosslinking mass spectrometry studies uncovered a multitude of direct Hsp90/Hsp70-ribosomal protein interactions[47,51]. In our studies, mutation of T492 had a substantial impact on Ssa1-ribosomal interactions and translational fidelity. Interestingly, T492 did not alter interactions with known orphan ribosomal proteins (oRPs). However, this was expected given that oRPs are regulated by Sis1[14], a co-chaperone that does not appear to be altered by T492 phosphorylation status in our studies. Heat shock causes global pausing of translational elongation[90]. This heat-mediated pausing is a response to Hsp70 dissociation with ribosomal machinery[32]. Our data show that the unphosphorylatable T492A variant remains locked on heavy polysomes during heat shock. This persistent binding is predicted to hinder elongation factor access, slow ribosome translocation, and reduce methionine-analog (HPG) incorporation, exactly as we observe. Importantly, polysome accumulation in the mutant is accompanied by slower ribosome run-off, indicating stalled, not active, ribosomes. Thus, higher polysome-bound Ssa1 does not equal higher productive translation. Minimally, these findings suggest that T492 phosphorylation contributes to heat shock-induced translational pausing to prevent the mistranslation or misfolding of newly synthesized proteins. Fascinatingly, the equivalent site in human

Hsc70 (T495) is phosphorylated by the Legionella LegK4 kinase and can lead to altered mammalian translation, suggesting an evolutionary conservation of function[66].

Stress granules (SGs) and processing bodies (P-bodies) are RNA structures in the cytoplasm formed during mRNA processing. Although they share some mRNA and protein components, they serve different cellular functions[91]. Processing bodies (PBs) maintain a baseline presence in unstressed yeast cells while showing dramatic induction under various stress conditions[24]. Interestingly, stress granules (SGs) emerge from existing PBs, suggesting that PBs serve as initial storage sites for messenger ribonucleoprotein complexes (mRNPs). The interactions necessary for stress granule or P-body formation are regulated by chaperones, specifically, Hsp70 and Hsp40[24]. Studies also have shown that defects in Hsp70 or Hsp40 function inhibited the disassembly of stress granules during the post-stress recovery phase but not the P-body[92]. While the importance of Hsp70 phosphorylation in RNA granule dynamics remains unexplored, this relationship warrants investigation. Our interactome data identified important proteins (Pab1 and Edc3) that are involved in biomolecular condensate formation. We studied the foci formation and the localization of Edc3-mCherry (P-body marker) and Pab1-GFP (stress granule marker) in WT and T492A cells under unstressed heat shock and recovery conditions. A striking observation was seen during the recovery time points. The mutant T492A was defective in Edc3-foci disassembly even at 2 h of recovery. However, the disassembly of stress granules was seen to be normal. This suggests that Ssa1 T492 phosphorylation is required for P-body disassembly. Our interactome analysis corroborated the localization data, revealing a preference for Edc3 to bind the WT vs T492A Ssa1, while no preference was observed for Pab1. The indifference of Pab1 to T492 status is intriguing, given its well-characterized interaction and regulation of the chaperone system[93–95]. Given the large number of PTMs present on Ssa1 and its associated co-chaperones, it is possible that other PTMs regulate Pab1-chaperone interactions. T492 lies in the substrate-binding domain of Ssa1, and in our mass spectrometry data, there were a large number of client interactions (approximately 40%) that were impacted by the mutation of T492. That the majority of client interactions remained unaffected overall implies that phosphorylation of T492 subtly fine-tunes client binding as opposed to a binary on-off switch. In-depth proteomic analysis of direct Hsp70 interactions in cells using proximity labeling and crosslinking mass spectrometry have revealed that Hsp70 has a preference for intrinsically disordered domains[47,96]. Interestingly, a similar phenomenon has been observed for yeast Hsp90[49]. However, despite a clear preference for Ssa1 to bind IDPs, our mass spectrometry experiment revealed that these tended to be independent of T492 phosphorylation status.

T492 is highly conserved throughout the Hsp70 family of proteins, it is present in all eukaryotic cells. It is due to this conservation and location in the client-binding domain that it was originally identified as part of a PTM "hotspot" with potential functional importance[97]. In yeast, the Hsp70 isoforms Ssa1, 2, 3 and 4 are partially redundant, and any single Ssa can support viability in the absence of the other three[3]. Although Ssa1 and 2 are constitutively expressed, Ssa3 and 4 are heat-induced, presumably to cope with the increased unfolded protein load. For the first tim,e we demonstrate that all four Ssa1 isoforms can be rapidly phosphorylated on T492 under heat stress. Interestingly, this suggests that Ssa3 and 4 may only ever be present in a T492-phosphorylated state when cells are exposed to heat stress. In addition to cytosolic chaperones, cells express a range of organelle-specific Hsp70s such as the ER-localized Hsp70 BiP/Grp78[1,98]. Although phosphorylation of BiP has not been observed at T518[99] (the equivalent site to T492 in BiP), the same residue is AMPy-lated by the FICD enzyme[97]. AMPylation of BiP T518 enhanced peptide dissociation from BiP 6-fold and slowed ATP hydrolysis[100,101]. This raises several important questions for future study, including whether BiP

is phosphorylated at this site and whether phosphorylation and AMPylation work in opposition to regulate BiP activity.

Chaperone code research is still in its infancy. Despite the large number of PTM sites detected on Hsp70, less than ten have been fully characterized in terms of their role and regulation[1]. There are some similarities between T492 and another well-characterized Ssa1 phospho-site, T36. Although these sites are differentially regulated (T36 by nutrient availability, T492 by heat), their phosphorylation leads to similar co-chaperone interaction changes. Despite their distance apart on the Ssa1 protein, both the phosphorylation of T36 and T492 lead to loss of Ssa1-Ydj1 interaction while preserving interaction with Sis1[49].

This study demonstrates an aspect of the heat shock response that explains how cells can respond so rapidly by promoting Hsp70 phosphorylation. Importantly, it solves a long-standing question of how distinct features of the yeast shock response that include Hsf1 activation, CWI pathway activation, ribosomal rearrangement and heat-induced quinary complex resolving are integrated (Fig. 8). Going forward, it will be interesting to understand how distinct features of the HSR, such as chaperone titration and heat-mediated HSF conformational changes, are co-regulated throughout cells and tissues.

## Methods

### Reagents and resources
Details on all reagents and resources (yeast strains and plasmids), primer sequences are provided in Supplementary Data 1–5.

### Yeast Strains and growth conditions
Yeast cultures were grown in either YPD (1% yeast extract, 2% glucose, 2% peptone) or grown in SD (0.67% yeast nitrogen base without amino acids and carbohydrates, 2% glucose) supplemented with the appropriate nutrients to select for plasmids and tagged genes. *Escherichia coli* DH5α was used to propagate all plasmids. *E. coli* cells were cultured in Luria broth medium (1% Bacto tryptone, 0.5% Bacto yeast extract, 1% NaCl) and transformed to ampicillin or kanamycin resistance by standard methods. To construct the T492A yeast strain, we used a classic plasmid swap method[37]. The T492A mutation was engineered by Genscript into plasmid pC210, which drives *SSA1* expression from the constitutive *SSA2* promoter[102]. For FLAG-tagged versions of Ssa1 (WT and T492A) were created via Genscript by insertion of a N-terminal FLAG-tag into Ssa1-pC210. Full sequences of all these plasmids are available in Supplementary Data 5. WT or T492A Ssa1 constructs (either untagged or FLAG-tagged) were transformed into yeast strain[37] ssa1−4Δ using PEG/lithium acetate. After restreaking onto media lacking leucine and 5-fluoro-orotic acid (5-FOA), resulting in yeast that expressed WT or T492A Ssa1 (tagged or untagged) as the sole cytoplasmic Hsp70 in the cell.

For epitope tagging the genomic copy of *HSP12* with a GFP epitope tag at the carboxy terminus, the pFA6a-GFP-His3MX6 plasmid was used. For epitope tagging the genomic copy of *MKK1, and MKK2* with an HA epitope tag at the carboxy terminus, the pFA6a-3HA-His3MX6 plasmid was used. Primer sequences are available in Supplementary Data 4. Genotypes were confirmed by PCR analysis.

To generate a strain for the inducible degradation of Pkc1, the OsTir1-Myc tagged gene was first integrated into the yeast genome at the *URA3* locus from ADHIp-OsTIR1-9Myc plasmid via PCR-mediated integrated gene tagging. Correct integration was confirmed via PCR. *PKC1* was tagged with the degron domain via PCR-mediated integrated gene tagging. The C-terminal degron cassette was generated by amplifying it from the plasmid pMK43 (IAA17−KanMX4). This cassette was then transformed into yeast strains expressing the *Oryza sativa* F-box protein, OsTIR1. Cells that grew on media supplemented with G418 (Kanamycin) were restreaked and checked via PCR to confirm degron tag integration. Functionality of the Pkc1-AID system was verified by examining cell viability on media containing or lacking auxin. Details of the yeast strains used are provided in Supplementary Data 2.

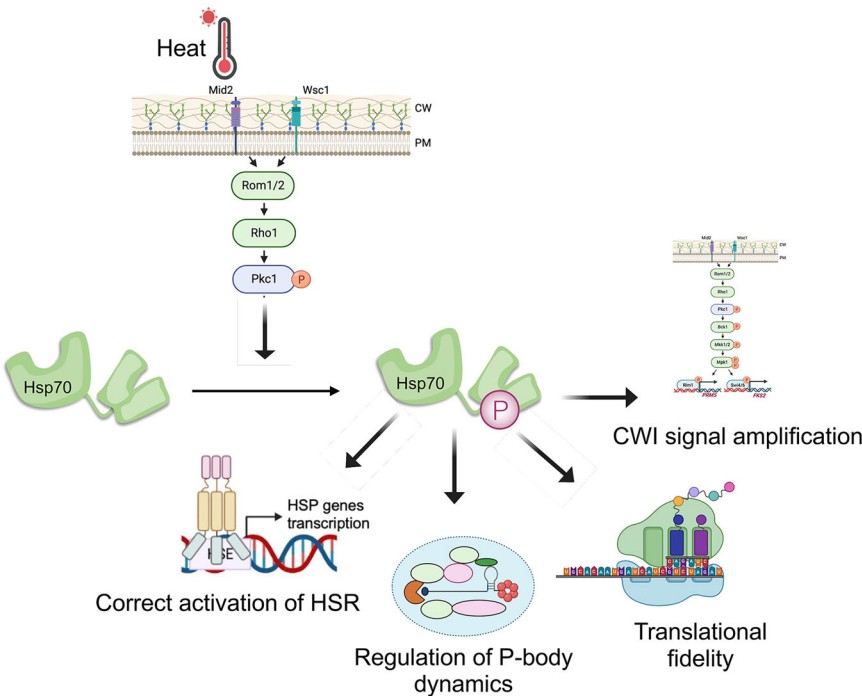

**Fig. 8 | Summary figure.** Pkc1-mediated phosphorylation of yeast Hsp70 regulates multiple aspects of the response to heat shock. Created in BioRender. Truman, A. (2025) https://BioRender.com/fqsruff.

## Plasmid construction

Plasmids for the expression of Bck1 and yeast Hsp70s were synthesized by Vectorbuilder. Mutants of Ssa1 were created by Genscript. Full sequences of these plasmids are available in Supplementary Data 5. Details of the plasmids used are provided in Supplementary Data 2.

## Growth assays

For serial dilutions, cells were grown to mid-log phase, 10-fold serially diluted and then plated onto appropriate media using a 48-pin replica-plating tool. Images of plates were taken after 3 days at 30 °C unless specified.

## Pkc1-AID experiments

The Pkc1-AID strain was grown to the mid-log phase at 25 °C. 1 mM auxin was added to ablate the Pkc1 function for the indicated time points, and then the yeast were shifted to 39 °C for 30 min.

## Luciferase assay

For the real-time luciferase activity assay, cells expressing the pHSE-lucCP+ plasmid[11] were grown to mid-log phase at 25 °C. Activity of Hsf1 was determined by adding luciferin (final concentration 0.5 mM) and distributing 150 μl aliquots of the cultures into a white 96-well plate. Cells were incubated in a Synergy MX Microplate reader (Biotek Instruments) at 39 °C for 200 min, and luminescence was read every 5 min. The graph was prepared using GraphPad Prism 7. All experiments were conducted with at least 5 biological replicates.

## Immunoprecipitation

For FLAG IP, cells were harvested, and FLAG-tagged proteins were isolated as follows: Protein was extracted via bead beating in 500 μl binding buffer (50 mM Na-phosphate, pH 8.0, 300 mM NaCl, 0.01% Tween-20). 200 μg of protein extract was incubated with 30 μl anti-Flag M2 magnetic beads (Sigma) at 4 °C overnight. Anti-FLAG M2 beads were collected by magnet then washed 5 times with 500 μl binding buffer. After the final wash, the buffer was aspirated, and beads were incubated with 65 μl Elution buffer (binding buffer supplemented with

10 μg/ml 3X FLAG peptide (Apex Bio) for 1 h at room temperature, then beads were collected via magnet. The supernatant containing purified FLAG-protein was transferred to a fresh tube, 25 μl of 4X SDS-PAGE sample buffer was added and the sample was denatured for 5 min at 95 °C. The eluates were separated by SDS-PAGE (7.5–10%) and visualized by immunoblotting.

For HA IP, cells were harvested, and HA-tagged proteins were isolated as follows: Protein was extracted via bead beating in 500 μl binding buffer (50 mM Na-phosphate pH 8.0, 300 mM NaCl, 0.01% Tween-20). 200 μg of protein extract was incubated with 30 μl anti-HA magnetic beads (Thermo Fisher Scientific) at 4 °C for 30 min. Anti-HA beads were collected by magnet then washed 5 times with 500 μl binding buffer. After the final wash, the buffer was aspirated, and beads were incubated with 65 μl Elution buffer and 15 μl of 4X loading dye and boiled at 100 °C for 10 min, then beads were collected via magnet. The supernatant containing purified HA-protein was transferred to a fresh tube, 25 μl of 4X SDS-PAGE sample buffer was added and the sample was denatured for 5 min at 95 °C. The eluates were separated by SDS-PAGE (7.5–10%) and visualized by immunoblotting.

For GFP IP, cells were harvested, and GFP-tagged proteins were isolated as follows: Protein was extracted via bead beating in 500 μl binding buffer (50 mM Na-phosphate, pH 8.0, 300 mM NaCl, 0.01% Tween-20). 200 μg of protein extract was incubated with 30 μl anti-GFP magnetic beads at 4 °C overnight. Anti-GFP beads were collected by magnet, then washed 5 times with 500 μl binding buffer. After the final wash, the buffer was aspirated, and beads were incubated with 65 μl Elution buffer and 15 μl of 4X loading dye and boiled at 100 °C for 10 min, then beads were collected via magnet. The supernatant containing purified HA-protein was transferred to a fresh tube, 25 μl of 5x SDS-PAGE sample buffer was added, and the sample was denatured for 5 min at 95 °C. The eluates were separated by SDS-PAGE (7.5–10%) and visualized by immunoblotting.

## Immunoblotting

Protein extracts were made as described[41]. 15 μg of protein was separated by 4%–12% NuPAGE SDS-PAGE (Thermo). Proteins were detected

using the following antibodies; Anti- PhosphoT492 (21st-century bio-chemicals), Anti-PGK1 (Thermo #22C5D8), Anti FLAG (Sigma, #F1365), Anti-His (QIAGEN #34670), Anti-GAPDH (Thermo MA5-15738), Anti-Hsc70 (Rockland immunochemicals #200301F64), Anti-HA (Thermo #26183), Anti-Ydj1 (StressMarq #SMC-166D), Anti-Sis1(Gift from Dr. E Craig), Anti-Hsp26 (Gift from Dr. J Buchner), Anti-GFP (Roche #1814460), Anti-Mpk1 (SCBT #133189), Anti-mAID (Proteintech #28209-1-AP) and Anti-phospho Mpk1 (Cell-signaling #4695). Blots were imaged on a ChemiDoc MP imaging system (Bio-Rad) after being treated with Super Signal West Pico PLUS Chemiluminescent Substrate (Thermo). For multiplex blots, fluorescent secondary antibodies (Bio-Rad Starbright #12004159 and #12005867) were added to the blots and imaged on the ChemiDoc MP imaging system (Bio-Rad) using multiplex settings. The blots were then subsequently stripped and re-probed with relevant antibodies using a mild or harsh stripping buffer (glycine, SDS, Tween20, pH 2.2, or 10% SDS, 0.5 M Tris HCl, pH 6.8 with β-mercaptoethanol, respectively).

## In vitro kinase assays

Wild-type FLAG-Ssa1 were transformed with a plasmid expressing HA-Pkc1 under a galactose promoter[41]. Cells were grown in SD-URA Galactose at 25 °C overnight, and grown to the mid-log phase and heat shocked for 30 min. Protein was extracted via bead beating. For HA pulldown, anti-HA antibody-conjugated beads (Thermo Fisher Scientific) were washed 3 times with kinase buffer (100 mM Tris HCl (pH 8), 100 mM NaCl, 10 mM MgCl$_2$ and 20% glycerol). 100 µg of protein extract was added to the beads, which were incubated at 4 °C for 1 h. The supernatant was removed, and the beads were supplemented with 40 µl of kinase buffer along with 10 µg of purified full-length His-Ssa1 (gift from Dr. C Prodromou). After substrate addition, 100 µM of ATP was added to initiate the kinase reaction. The reaction was incubated for 1 h at room temperature. The reaction was terminated by adding 60 µl of 5 x SDS-PAGE sample buffer (loading buffer), followed by boiling for 10–15 minutes. Samples were then separated using 4–12% SDS-PAGE gel and immunoblotted using phospho T492 antibody, His antibody, and HA antibody.

## Pab1 and Edc3 foci quantification

Yeast cells from cultures grown to log phase (OD600 ≈ 0.5) were mounted in selective growth medium (C-URA), and 3D image stacks were collected at 0.3 µm z increments on a DeltaVision Elite Work-station (Cytiva) based on an inverted microscope (IX-70; Olympus) using a 100 × 1.4NA oil immersion lens. Images were captured at 22 °C with a 12-bit charge-coupled device camera (CoolSnap HQ; Photo-metrics) and deconvolved using the iterative-constrained algorithm and the measured point spread function. Image analysis and pre-paration was done using Softworx 6.5 (Cytivia) and FIJI ImageJ[103]. For heat shock experiments, cells were incubated in growth media in a 39 °C water bath (New Brunswick Scientific) for 15 min, 30 min, 1 hour, or 2 h, then immediately mounted on slides for imaging. To quantify the presence of Pab1 and Edc3 foci, a minimum of 100 cells were analyzed for each timepoint. Wildtype or T492A cells were visually scored for the presence of internal fluorescent foci from 3D projection images. Timepoints for the presence of foci were compared via single-factor ANOVA analysis. All experimental conditions were repeated in biological triplicate.

## Polysome profiling

To grow cells for polysome profiling, 200 mL of yeast were grown in SCD-Leu overnight with shaking at 25 °C until they reached an OD600 = 0.4. Cells were collected onto a 0.45 µM filter (Cytiva #60173) using vacuum filtration (Sigma #Z290432) and immediately trans-ferred to 100 mL of SCD-Leu prewarmed to 25 °C or 39 °C. After 10 minutes of incubation with periodic swirling the cells were either cooled back to 25 °C in a water bath and then transferred to a shaking

incubator at 25 °C or harvested immediately via vacuum filtration onto a filter, followed by scraping the cells (Fisher #8100241) and plunging directly into liquid N2. The frozen cells were transferred to a pre-chilled 2 mL tube (Eppendorf #022363352). Cells were lysed with a pre-chilled 7 mm stainless steel ball (Qiagen #69990) using four 90 sec, 30 Hz pulses in a Retsch MM 400 mixer mill, chilling in liquid N2 between pulses. Sample was resuspended in 900 µL polysome lysis buffer (20 mM HEPES-KOH pH 7.4 (Sigma #H4034), 100 mM KCl (Fisher #BP366), 5 mM MgCl2 (Sigma #M2670), 200 µg/mL heparin (Calbiochem #375095), 1% triton X-100 (Sigma #T8787), 0.5 mM TCEP (Goldbio #TCEP25), 100 µg/mL cycloheximide (Thermo #J67043AD), 40 U/ml RNase inhibitor (NEB #M0314L), 1:100 EDTA-free Halt pro-tease inhibitor (Thermo #PI78437). The lysate was clarified by cen-trifugation at 3000 x $g$ for 2 min, and the clarified lysate was transferred to a new tube. The OD260 of each lysate was measured using an LVis plate on a CLARIOstar Plus (BMG), and the concentra-tions were normalized across samples using lysis buffer. Finally, ali-quots were flash frozen in liquid N2, and total samples were collected by mixing 20 µL of sample with 2x LDS buffer (500 mM Tris pH 8.5, 4% lithium dodecylsulfate, 20% glycerol, 1 mM EDTA, 0.2%(w/v) orangeG, 2.5% 2-Mercaptoethanol). A 10–50% continuous sucrose gradient in polysome gradient buffer (5 mM HEPES-KOH pH 7.4, 140 mM KCl, 5 mM MgCl$_2$, 100 µg/ml cycloheximide, 0.5 mM TCEP) was prepared in SW 41Ti tubes (Seton #7030) using a Biocomp Gradient Master with the 10–50% short sucrose program and cooled to 4 °C. Clarified lysate (200 µL) was loaded on top of the gradient, and gradients were spun in a SW41Ti rotor at 281000 x $g$ for 105 min at 4 °C. Gradients were fractionated into 0.8 mL fractions using a Biocomp Piston Gradient Fractionator with a Triax flow cell monitoring at 260 nm. Polysome to monosome ratios were calculated using background-subtracted values. Proteins were precipitated from fractions using Tri-chloroacetic acid (TCA). First, 5 µL of 1 mg/mL BSA was added to each fraction as carrier protein. Then TCA was added to a final concentra-tion of 10%, samples were vortexed and frozen at − 20 °C for at least one hour. Samples were then centrifuged at 4 °C, 21,000 x $g$ for 15 min, and the supernatant was removed. Fractions were combined together by centrifugation in the same tube before a final wash with 10% TCA. Each tube was then washed twice with chilled acetone, dried for 5 min at 70 °C before being resuspended in 2 x LDS sample buffer (500 mM Tris pH 8.5, 4% lithium dodecylsulfate, 20% glycerol, 1 mM EDTA, 0.2% (w/v) orangeG, 2.5% 2-Mercaptoethanol), incubated with shaking at 70 °C for 10 minutes, vortexed and finally centrifuged at 21,000 × $g$ for 1 min. Samples were then run on a Bis-Tris, 4–12% gel with MES buffer (Genscript #M00654), transferred to PVDF membrane (MilliporeSigma #IPFL85R) using a Trans-blot turbo transfer system (Bio-rad #1704150) in transfer buffer (0.3 M glycine, 0.3 M Tris, 0.1% SDS, 20% methanol). Membranes were blocked with 5% milk for 1 hr before incubating with the specified primary antibodies overnight. Westerns were then visualized using donkey anti-mouse 800CW (LI-COR #925-32212) and goat anti-rabbit 680 (LI-COR #925-68071) on a LI-COR Odyssey M imager.

## Translation activity via HPG incorporation

Translation was measured via HPG incorporation, based off the method used in refs. 68,104,105 Yeast cells were grown to OD600 = 0.3 in SCD-leu medium (Sunrise #1304-030) at 25 °C. Cells were trans-ferred to a 2 mL deep well plate (VWR #75870-792), pelleted by cen-trifugation at 2000 x $g$ for 5 min and resuspended in SCD-met (Sunrise #1343-030). For cycloheximide (CHX) treated samples, CHX (Thermo Scientific #357420010) was also included at 100 µg/mL. Cells were pelleted again and resuspended in 900 µL pre-warmed SC-met con-taining 50 µM of homopropargylglycine (HPG, Vector Laboratories #CCT-1067-25) and CHX for controls or in SC-met with no HPG. Cells were incubated at the indicated temperature (25 °C or 39 °C) for the specified time points (10, 20, 30 min) in a waterbath. Protein synthesis

was halted by the addition of 100 μL of 1 mg/mL CHX. Cells were then pelleted, resuspended with 300 μL of ice-cold PBS + 100 μg/mL CHX, and fixed by the addition of 700 μL of ice-cold ethanol. Cells were fixed for 1 hr at RT or overnight at 4 °C. They were then washed with PBS + 3% BSA (Gold Bio #A-420-100), and resuspended in click chemistry reaction buffer (50 mM Tris pH 7.5, 150 mM NaCl, 2.5 mM THPTA (Combi-blocks #QH-3278-1g), 1 mM CuSO$_4$ (Sigma #C1297100G), 2.8 mM fresh sodium ascorbate, and 5 μM AFDye 647 Picolyl Azide (Vector #CCT13001)). After 30 minutes, cells were pelleted, washed once with 3% BSA in PBS, once with wash buffer (0.5 mM EDTA, 2 mM NaN$_3$), and finally resuspended in PBS. Samples were measured using a Cytek Aurora 4 flow cytometer at the Texas A&M University Flow Cytometry Facility. The data were then analyzed in R using custom code utilizing the flowCore (https://doi.org/10.1186/1471-2105-10-106), tidyverse (https://doi.org/10.21105/joss.01686), and sf package (https://doi.org/10.32614/RJ-2018-009). Singlets were selected based on FSC.A, SSC.A and FSC.H values. The brightest 647 channel (R2) was then normalized to the SSC.A channel for every cell to control for cell size. Cells were filtered for those with higher intensities than the no HPG control. The mean normalized 647 intensity of the population was then calculated for every sample. Finally, this mean intensity was normalized to the mean of the WT samples at 10 minutes at a given temperature. At least 1000 cells were quantified in every sample.

## Mass spectrometry

Six immunoprecipitations (three biological replicates of wildtype Ssa1 controls and three biological replicates of Ssa1-T492A phosphodeficient mutants) were eluted from beads using 8 M urea, 10 mM DTT in 50 mM Tris pH 8.5 for 45 min at room temperature with mixing at 600 rpm before alkylation with 50 mM IAA for 30 min in the dark. Samples were diluted sixfold with 50 mM Tris pH 8.5 to reach < 2 M Urea concentration and digested with 0.4 μg of trypsin-LysC mix (Promega) overnight at 37 °C. Tryptic peptides were desalted with Pierce C18 Desalting Spin Columns (Thermo Fisher Scientific) according to the manufacturer's protocol, dried down on SpeedVac, and resuspended in mobile phase A (0.2% formic acid in water) immediately prior to mass spectrometric analysis.

## Liquid chromatography – tandem mass spectrometry peptide analysis

Resuspended peptides were separated by nanoflow reversed-phase liquid chromatography (LC). An Ultimate 3000 UHPLC (Thermo Scientific) was used to load ~1 μg of peptides on the column and separate them at a flow rate of 300 nL/min. Each sample was injected twice to allow for the use of different fragmentation techniques. The column was a 15 cm long EASY-Spray C18 (packed with 2 μm PepMap C18 particles, 75 μm internal diameter, Thermo Scientific). The analytical gradient was performed by increasing the relative concentration of mobile phase B (0.2% formic acid, 4.8% water in acetonitrile) the following steps: from 2% to 30% in 32 min, from 30% to 50% in 5 min, and from 50 to 85% in 5 min (for washing the column). The 4 min wash at high organic concentration was followed by moving to 15% in 2 minutes, increasing to 70% in 1 min for a secondary wash before re-equilibration of the column at 2% B for 7.5 min, for a total run time of 68 min. A 2.2 kV potential was applied to the column outlet using an in-house nanoESI source based on the University of Washington design for generating nano-electrospray.

All mass spectrometry (MS) measurements were performed on a tribrid Orbitrap Eclipse (Thermo Scientific). Broadband mass spectra (MS1) were recorded in the Orbitrap over a 375–1500 *m/z* window, using a resolving power of 120,000 (at 200 *m/z*) and an automatic gain control (AGC) target of 4e5 charges (maximum injection time: 50 ms). Precursor ions were quadrupole selected (isolation window: 1.6 *m/z*) based on a data-dependent logic, using a maximum duty cycle time of 3 s. Monoisotopic precursor selection and dynamic exclusion (60 s)

were applied. Peptides were filtered by intensity and charge state, allowing the fragmentation only of precursors from 2 + to 7 + . Tandem mass spectrometry (MS2) was performed by fragmenting each precursor passing the selection criteria using both higher energy collisional dissociation (HCD; first injection) with normalized collision energy (NCE) set at 30% and electron transfer dissociation – higher energy collisional dissociation (EThcD; second injection), with ETD reagent target set at 5e5, reaction time calculated on the basis of a calibration curve and supplemental collisional activation set at NCE = 10%. The AGC target for both HCD and EThcD MS2 was set at 8e4 (maximum injection time: 55 ms), and spectra were recorded at 15,000 resolving power. A list of all interactors identified is provided in Supplementary Table 3.

## MS Data analysis

For general identification of all proteins included in the samples, HCD fragmentation data were processed with Protein Discover 2.4, utilizing Sequest HT and MS Amanda search engines. For both, Precursor Mass Tolerance was 10 ppm, and Fragment Mass Tolerance was 0.02 Da. Carbamidomethylation (C) was allowed as a static modification, and dynamic modifications were as follows: Oxidation(M), Acetyl (protein N-term). Identified peptides were validated using Percolator, and the target FDR value was set to 0.01 (strict) and 0.05 (relaxed). Finally, results were filtered for high-confidence peptides using consensus steps. Control peptide error rate strategy was used, and 0.01 (strict) and 0.05 (relaxed) values were used for Target FDR for both PSM and Peptide levels. Changes in protein abundance between the two strains were statistically tested by ANOVA using the built-in function within Proteome Discoverer. 3 replicates were used for each strain. The data was normalized on Ssa1.

## Gene ontology enrichment analysis

Gene ontology analysis of Ssa1 immunoprecipitated complexes were accomplished using Yeast GO Slim.

## Calculation of protein intrinsically-disordered content

The percentage of intrinsically disordered residues for interactors that are preferentially bound to WT or T492 Ssa1 were obtained via the MobiDB database (https://mobidb.org/). Proteins were considered disordered if they contained > 40% disordered residues.

**General data Analyses.** Data processing and analyses were performed using GraphPad Prism (version 7).

## Quantitative RT-PCR

Quantitation of yeast SSA1 and Luciferase transcription was carried out as in ref. 106. Briefly, yeast cells were grown overnight at 30 °C, re-inoculated at OD600 of 0.2–0.4 and then grown for a further 4 h at 25 °C. Cells were heat shocked at 39 °C for 30 min, and total RNA was extracted from cells using a GeneJet RNA extraction kit. As a control, untreated cells were left at 25 °C. Total RNA (1 μg) was treated with 10 units of RNase-free DNase I (Thermo) for 30 min at 37 °C to remove contaminating DNA. DNAse I activity was stopped by adding 1 μL of 50 mM EDTA and incubating at 65 °C for 10 min. cDNA synthesis was carried out by iScript reverse transcriptase (BioRad) on aliquots of 1 μg RNA. The single-stranded cDNA products were used in qPCR on an ABI Fast 2000 real-time PCR detection system based on SYBR Green fluorescence. Sequences of oligo pairs are listed in Supplementary Data 4. Expression was normalized against that of *ACT1* in each strain, and the resulting ratios in WT cells were defined as one-fold.

## Reporting summary

Further information on research design is available in the Nature Portfolio Reporting Summary linked to this article.

## Data availability

The raw proteomic data can be found on the MassIVE database (MSV000096492 [https://massive.ucsd.edu/ProteoSAFe/dataset.jsp?accession=MSV000096492]) and cross-listed on ProteomeXchange (PXD058151 [http://proteomecentral.proteomexchange.org/cgi/GetDatase t?ID = PXD058151]). The list of yeast strains used are included in Supplementary Data 1. List of plasmids used is included in Supplementary Data 2. List of interactors identified from MS are included in Supplementary Data 3. List of primers in included in Supplementary Data 4. Plasmid sequencing data included in Supplementary Data 5. Source data are provided in this paper.

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

## Acknowledgements

The authors are grateful to Dr. D Levin, Dr. S. Kron, Dr. R Parker, Dr. K Morano, Dr. E Craig, Dr. C. Prodromou, Dr. C. Andréasson and Dr. B. Freeman for providing materials used in this study. This work was supported by the NIH (R21NS133682 and R01GM149639 to AWT; R35GM147397 to L.F.). Use of the Texas A&M University Flow Cytometry Facility (RRID:SCR_022169) is acknowledged. This work was supported by the National Science Foundation 2028519 to R.J.C.

## Author contributions

Conceptualization: A.W.T, Data curation: S.O, J.T.K, J.H.G, D.S., and J.A.B Software: S.O, J.T.K., and J.H.G Formal analysis: S.O, J.T.K, J.H.G, D.S, J.A.B., and A.W.T Investigation: S.O, R.J.C, L.F., and A.W.T Writing—original draft: S.O, A.W.T, J.T.K J.H.G, R.J.C, L.F., and J.A.B writing—review and editing: all authors; supervision: funding acquisition: A.W.T.

## Competing interests

The authors declare no competing interests
