## [Transparent Peer Review file · Nature Communications]

Mechanosensor-mediated Hsp70 phosphorylation orchestrates the landscape of the heat shock response

Corresponding Author: Dr Andrew Truman

A version of this paper was originally rejected for publication by Nature Communications, however that decision was reconsidered after appeal by the authors.

Version 1:

Reviewer comments:

Reviewer #1

(Remarks to the Author)

In the manuscript by Omkar et al., the main claim is that heat shock mediates the phosphorylation of Hsp70 at T492 and this phosphorylation is responsible for a wide variety of effects of the heat shock response. They demonstrate that heat shock induces rapid phosphorylation of Ssa1 T492. Interestingly, this effect is independent of protein unfolding but dependent on the mechanosensor Mid2 and is exerted directly by the protein kinase C (Pkc1). They show that T492 is critical for the activation of the heat shock response and claim a role in the phosphorylation of this residue for HSF1-Ssa1 dissociation, protein translation, ribosome association, P-body disassembly, and CWI signaling. However, as detailed below, there are many flaws in the design, interpretation, and conclusions of the results that make this manuscript unsuitable for publication (the comments are ordered according to the order of results in the text):

- 1) The authors show that T492 phosphorylation of Ssa1 by heat shock depends on Mid2 but not Wsc1. Previous reports on the CWI sensors required for Mpk1 activation under heat shock conditions showed that both proteins (Mid2 and Wsc1) sense heat shock (Verna et al, PNAS, 1997; Gray et al, EMBO J, 1997; Ketela et al., J. Bacteriology, 1999). How do explain the authors these differences?
- 2) Regarding the experiments showing that Pkc1 mediates phosphorylation of Ssa1 at T492:
 - a) Figure S1 shows the cells' phenotype as validation of the Pkc1-AID strain, but levels of Pkc1 parallel to Phospho T492 (Figure 2d) need to be shown.
 - b) In the in vitro kinase assay (Figure 2f), Pkc1 was purified from cells growing at 25°C, but the effect of temperature (39°C) also needed to be assayed.
- 3) Regarding the Mass spec experiments comparing heat-shocked WT and T492A cells, I have several questions and comments (see below), but my main criticism is that it does not contribute much to the characterization of the functional role of Ssa1 T492 phosphorylation.
 - a) Although the objective is to compare and quantify differences in protein interactors of both proteins, using controls of untagged strains should avoid non-specific interactions.
 - b) Details about how quantification was measured are missing: How many replicates were analyzed? What kind of normalization was applied? Moreover, no experimental validation of any of the proteins identified was carried out. This is absolutely necessary, particularly in relation to some of the proteins whose functional relationship with Ssa1 phosphorylation is discussed throughout the work.
 - c) In 784: fragment mass tolerance (0.2 Da) is too high (an adequate setting should be 0.02).
 - d) 647 proteins are identified and 560 quantified, but only 380 are included in the Excel file. Why?
 - e) The text through lines 256-262 has several errors: the mutation does not impact the association of fifteen chaperones and co-chaperones, but only nine. The other six do not show differences.
 - f) One of the conclusions of the proteomic approach is that there are differences in the sets of client proteins bound to Ssa1 WT and T492A regarding the IDR composition of interactors. However, the authors do not clarify much about this conclusion. In line 251, they say that there are subtle differences; in the discussion (line 490), they say that it is possible that phosphorylation of T492 under heat shock promotes reinforced interaction IDP proteins, but in the abstract (ln 31), they

assert that the phosphorylation of T492 leads altered preference for clients with IDRs. Furthermore, there is no statistical analysis of the data shown in Figure 3e, which would be necessary in order to draw valid conclusions. In addition, the figure legend does not provide a clear description of what exactly is represented on the graph (mean value, etc.). Additionally, it appears that there are some negative values for the percentage of IDR in the client proteins; how can this be possible?

4) From the results of experiments described in Figure 4d, the authors claim that T492 phosphorylation is important for Ssa1-Hsf1 dissociation upon heat shock. This is a critical point in explaining the mechanism by which phosphorylation of Hsp70 by Pkc1 would trigger Hsf1-Hsp70 dissociation, leading to the activation of the HSF response. However, I do not think the results properly support this conclusion derived from the fact that mutation of T492 leads to the overexpression of Ssa1. Apparently, there are no differences in the amount of Ssa1 or Ssa1-T492A co-immunoprecipitated with Hsf1, but we can not exclude that this is due to the high amount of Ssa1 T492A and not to differences in the affinity between Hsf1 and Ssa1 as a consequence of the mutation. Experiments to measure differences in the affinity of Ssa1 variants for Hsf1 are needed. Additionally, reducing the amount of extract in the Co-IP and co-IP experiments in the opposite direction (immunoprecipitation with anti-Flag) could help. Moreover, Co-IP experiments should include a control of cells expressing only GFP to exclude possible nonspecific binding. Additionally, experiments showing that mutation of Thr 492 to Glu or Asp reverses the loss of dissociation should be appropriate. Contrary to Figure 4d's conclusion, the MS interactors results provided in Table S3 show that Hsf1 did not display a preference for Ssa1 T492A ($\text{Log}_2(\text{WT}/\text{T492A})=-0.075$), but this is not discussed.

5) The authors argue (In 300-306) that overexpression of Ssa1 T492A is due to an activation of a Msn2/4 transcriptional response, but from my point of view, this is an overestimated conclusion. The authors show that levels of Hsp12 are elevated in these circumstances, but no additional proof (i.e mRNA levels of SSA1 in a msn2/4 mutant strain; activation of Msn2/4; nuclear translocation) is shown. Moreover, the authors claim that their data demonstrate that T492 phosphorylation is important for the activation of both Hsf1 and Msn2/4-mediated heat shock-induced transcriptional response. However, levels of Hsp12 in the T492A strain are induced at 25°C, suggesting that Msn2/4 activation in this strain, if it exists, is not mediated by heat shock but induced constitutively. It is also important to mention here that, in contrast to these results, previous works (Beltrao et al, Cell. 2012 Jul 20;150(2):413-25) did not report overexpression of Ssa1 in a T492A S495A strain. This is note even commented in the manuscript.

6) Regarding the results relating Hsp70 phosphorylation and ribosome association:

- a) Conclusions from quantifying data from the Western blots in Figure 5d require further explanations about the number of replicates and statistical analysis of the differences. Otherwise, conclusions should be overestimated and not supported by the data.
- b) As stated by the authors (In 310), Hsp70 dissociation from the ribosomes during heat shock promotes translational pausing. However, the author's results do not fit with this argument since they observe that, while polysome-bound Ssa1-T492A increases with respect to the WT, the T492A strain displays lower translation rates than the WT. Moreover, other publications studying the effect of phosphorylation of this residue report opposite conclusions. Moss et al. (Cell Host Microbe, 2019) reported that phosphorylation of the human Hsp70 in the equivalent residue (T495) increases its binding to polysomes, leading to translation inhibition. Additionally, Beltrao et al (2012), showed that the mutant T492A S495A was defective in polysome binding.
- c) Looking at the graphics in Figure 5f, it seems that the threefold increase in the mutant referenced in the text (In 339) is overestimated. Additionally, at what temperature were these experiments performed? Is this effect a consequence of heat shock?
- d) In the legend of Figure 5f, they mention that statistical significance was calculated via ANOVA. However, they only have two groups, so this is not an appropriate analysis.

7) Regarding the experiments of P-body formation upon heat shock and disassembly:

- a) The description of these results is not acceptable. What was the heat shock temperature used in these experiments?. Looking at the legend of Figure 6, the experiments were carried out at 46°C for 15 min and 30 min. However, looking at the methods section they say that these experiments were carried out at 39°C. Moreover, heat shock recovery in Figure 6 is indicated at 30°C during 60 and 120 min, but in methods, they describe 1 or 2 hours of heat shock at 39°C. Stress granules in response to heat shock (46°C) have been described in several reports but at shorter times (10 min). Since heat shock experiments carried out by the authors throughout the whole paper have been developed at 39°C, these experiments should be also done at 39°C (15, 30 min) and 46°C (10 min)
- b) In addition to % of cells containing P-bodies, the number of P-bodies per 100 cells should be included.
- c) The effect of the T492A mutation in Pab1 foci disassembly shown in the graphics of Figure 6 does not correspond to the pictures: in the picture of Pab1/WT Ssa1 in recovery (30°C) there are almost no foci.
- d) How do the authors explain that in the majority of the pictures stress granules (Pab1) co-localize with P-bodies (Edc3)?

8) Regarding the relationship between heat-induced phosphorylation of Hsp70 and CWI pathway, the authors show that heat shock-mediated phosphorylation of Hsp70 in T492 via Pkc1 is absolutely necessary for the activation of Mpk1 and for the transcriptional response via the CWI pathway. This is quite interesting and innovative. However, the experiments developed to find out the mechanism responsible for this connection do not support the conclusions, and the mechanism is not characterized:

- a) The authors focus on the possibility that the effect of Ssa1 T492A mutation on Mpk1 activation can be a consequence of differences in the levels of the protein kinases Mkk1, Mkk12, and Bck1, but there are many other possibilities.
- b) Even focusing only on this possibility, the results do not support the conclusions:
 - They work with two fragments of Bck1 instead of the full-length protein because they were unsuccessful in detecting Bck1

even in WT cells (ln 373). Even they argue explicitly (ln 375) that since Bck1 discovery, there are no reports containing Western blots of this protein. However, this is not true: to my knowledge at least in two publications full-length versions of Bck1 fused to Myc (Hruby et al, J Cell Sci, 2011, 124 (1):35-46; Garcia et al, J Cell Sci, 2016, 129: 1649-1660) have been reported, so all the experiments using the Bck1 protein fragmented in two pieces (Figure 7 g, h, i) are not justified.

- Moreover, the authors describe in line 238 that co-IP experiments were done in WT and T492A strains but no CoIP of the mutant is shown in Figure 7
- In any case, the decrease in the levels reported in for the carboxy fragment can not account for the complete blocking of Mpk1 activation and lack of induction of FKS2 and PRM5.

9) There are many errors and an absence of information in the majority of the figure legends

- Figure 1a: The predicted structure of Ssa1 is shown, but no reference is provided for how or where this prediction was obtained.
- Figure 1d: The Western blot only shows WT cells, despite the figure legend mentioning both WT and T492A cells.
- Figure 1e: The legend states that heat shock was performed at the indicated time points, but it was only performed at a single time point (1 h), as mentioned in the Results section.
- Figure 2d: The legend states that cells were treated with auxin and incubated at 39°C for different times. However, the actual experiment involved auxin treatment for the indicated times, followed by a 30-minute incubation at 39°C for all samples.
- Figure 2f: the legend mentions the use of antisera to FLAG epitope, but the epitope shown in the figure is HIS.
- Figure 3e: In line 864, a verb is missing, making the conclusion unclear.
- Figure 4d: The legend does not mention the antisera used for the PGK1 epitope.
- Figure 4f: The legend mentions the use of antisera to PGK1 epitope, but the epitope shown in the figure is GAPDH.
- Figure 5f: The temperature at which the experiment was performed is not indicated. Additionally, the legend mentions an ANOVA analysis, which is not appropriate since only two groups are being compared.
- Figure 7i: The legend states that "the C-terminus of Bck1 interacts with Bck1," but it should be "the C-terminus of Bck1 interacts with Ssa1."
- Figure S1: It needs further explanation

10) Regarding the "Materials and Methods" section, I found several inconsistencies and a lack of information necessary for experimental reproducibility.

- It is stated that reagents are provided in the supplementary tables, but this table is missing.
- The description of plasmid construction for the expression of Flag-Ssa2, Flag-Ssa3, Flag-Ssa4, and the HA-tagged Bck1 fragments lacks necessary details.
- The luciferase assay is described twice in this section, with inconsistencies between the two versions.
- In the first description (line 579), cells were grown at 25°C, while in the second (line 648), they were grown at 30°C.
- The first version states that incubation lasted 90 minutes with measurements taken every 5 minutes, whereas the second version states that incubation lasted 90 minutes with readings every 3 minutes.
- Both descriptions mention incubation at 37°C, but the correct temperature is 39°C according to the graph.
- In vitro kinase assay: The composition of the kinase buffer and loading buffer is not specified.
- Pab1 and Edc1 foci quantification. According to the figure, cells were heat-shocked at 46°C for 15 and 30 minutes, followed by recovery at 30°C for 1 and 2 hours. However, the Materials and Methods section states that cells were incubated at 39°C for 15, 30, 60, or 120 minutes, which does not match the figure.

Reviewer #2

(Remarks to the Author)

Reviewer #3

(Remarks to the Author)

This manuscript investigates the role of threonine 492 phosphorylation of yeast Hsp70 (Ssa1) during heat stress. The authors demonstrate that heat shock induces Ssa1 T492 phosphorylation, dependent on the cell wall sensor Mid2 and the Pkc1 kinase of the cell wall integrity (CWI) pathway. This phosphorylation is shown to be essential for Hsf1 release and proper activation of the heat shock response (HSR), potentially contributing to its rapid kinetics. Furthermore, T492 phosphorylation influences Ssa1's ribosome association and translation fidelity during heat stress, and regulates P-body disassembly during recovery. Notably, the CWI pathway is not activated in T492A mutant cells during heat shock, though a direct mechanistic link remains to be established.

This study demonstrates a novel and extremely intriguing connection between cell-wall integrity pathway-mediated Hsp70 phosphorylation and regulation of the heat shock response. It elucidates how environmental stress signals are transmitted from cell surface sensors to the nucleus, activating the stress response, thereby contributing to our understanding of the chaperone code. Overall, much of the data are very compelling and the quality of the results is high. However, open questions remain, and some of the results require additional support, as detailed below.

1. Please specify the heat shock duration for all the related Figures or at least mention it in the figure legends.
2. Figures 1c, 1d, 1e and 2e indicate a low level of T492 phosphorylation without heat shock. This should be explicitly

recognized in the text. The authors should also discuss potential kinases and the functional implications of this basal phosphorylation.

3. In Figure 2b, residual T492 phosphorylation is observed in the absence of Mid2. To ensure reproducibility of this finding, the authors should repeat this experiment a minimum of three independent times and present the average signal, similar to that in Figures 4c and 4f, or clarify if this was indeed done. Additionally, given the potential for delayed responses, this experiment should be conducted using the 4-hour timeline employed in Figure 1c to definitively rule out any delayed phosphorylation kinetics. The authors should also discuss why in the absence of Mid2, Wsc1 could not activate Pkc1 to phosphorylate Ssa1. It is unclear what significance should be attached to the apparent Mid2 specificity, if any.

4. In Figure 2d, the authors should examine Pkc1 protein levels upon auxin induction at the indicated time points to strengthen the correlation between T492 phosphorylation reduction and Pkc1 levels. This could be achieved by using anti AID tag antibody or tagging Pkc1 with additional small epitope tag, such as FLAG or HA. Furthermore, if generating a pkc1 null mutant in sorbitol-containing media is feasible, utilizing this mutant directly would provide a more straightforward and definitive assessment of Pkc1's role in T492 phosphorylation.

5. While Figure 2f demonstrates that Pkc1 can phosphorylate Ssa1 *in vitro*, given that T492 is within the preferred PKC substrate motif, *in vivo* validation of this interaction is important. To confirm this interaction within cells, a crosslinking and pulldown experiment using Pkc1 as bait to detect Ssa1 association and T492 phosphorylation should be considered. Notably, Pkc1 was absent from the Ssa1 interactors identified by mass spectrometry (MS) (Supplementary table S1). This discrepancy requires discussion within the text.

6. The authors noted that Ssa1 T492A exhibits significantly higher expression (at least five-fold) compared to Ssa1 WT (Figures 1c, 2c, 4d). In Figure 3, where MS data and changes in Ssa1 interactors are presented, it is unclear whether the levels of detected interactors were normalized to the respective levels of Ssa1 WT and Ssa1 T492A. If normalization was not performed, the observed preferential binding of interactors requires re-evaluation to account for the differential expression of Ssa1 variants.

7. The observed growth defect of the T492A mutant at 37 °C (Figure 4a) and the corresponding lack of Hsf1-mediated heat shock response (Figures 4b, c) are very intriguing. However, given the significantly higher expression of T492A compared to WT (at least five-fold), it is possible that the observed Hsf1 trapping is an artifact of T492A overexpression, rather than a direct effect of the T492A mutation itself. To address this, we recommend reducing T492A expression using a weaker heterologous promoter and examining whether Hsf1 still fails to be released under heat shock condition. Alternatively, a phosphorylation mimetic mutation (T492D) could be introduced to assess whether it retains the ability to bind Hsf1 upon heat shock.

8. Additionally, regarding Figure 4A, the observed temperature sensitivity at 37°C appears to be significantly remediated by the addition of sorbitol, an osmotic stabilizer. This finding suggests that the growth defect is not due to a reduced heat shock response, as mentioned in the text, but mostly due to the clear defects in CWI signaling shown in Figure 7. This in some way alters the interpretation of the effects of T492 phosphorylation in both the results and discussion and should be clearly explained.

9. In Figure 4d, the signal for WT Ssa1 appears relatively weak. To ensure the reliability of the observed reduction in Ssa1-Hsf1 binding upon heat shock, we recommend repeating this experiment at least two additional times and quantifying the average signal. Furthermore, overexpressing WT Ssa1 may enhance the signal and provide a more robust demonstration of the reduction in Hsf1 binding during heat shock.

10. The observation of Msn2/4 activation in the T492A mutant under normal conditions (Figure 4f) suggests a complete blockage of Hsf1 activity. This raises crucial questions regarding the mechanism of T492A's effect. Specifically, it is necessary to determine whether T492A expression is upregulated by the activated Msn2/4 pathway or the T492A mutation stabilizes the Ssa1 structure, thereby enhancing its apparent Hsf1 binding affinity.

11. In Figure 6e, the authors must rule out whether the observed slow clearance of P-bodies in T492A cells is a consequence of the increased abundance of Ssa1 T492A. This could be done using a lower expression construct as described above.

12. The observed inactivation of the CWI pathway in the T492A mutant (Figure 7a-c) is an interesting finding. However, the *in vivo* interaction of Ssa1 with Bck1 fragments (Figure 7h, i) raises a significant discrepancy, as Bck1 was not identified as an Ssa1 interactor in the mass spectrometry data (Supplementary table S1). The authors must address this discrepancy and clarify whether the Bck1 fragments represent the native state of Bck1 or potentially misfolded states that attract Ssa1 binding, taking into consideration the challenges inherent with working with the full-length protein. Furthermore, the phosphorylation status of Mkk1/Mkk2 in T492A cells under heat shock condition should be evaluated to further elucidate the CWI pathway inactivation.

Minor comments:

13. Introduce the full terms for AZC and CPZ upon their first mention in the text.

14. Add periods at the end of sentences on lines 99, 167, 467, and 716.

15. Use subscript for numerical values in chemical formulas (e.g., NaN₃, N₂, CuSO₄, MgCl₂).

16. Common chemical names, such as 'glycerol,' should not be capitalized unless they appear at the beginning of a sentence.

17. Line 685: Remove the extra ")".

18. Line 308: The title 'Hsp70 phosphorylation is critical for protein translation and ribosome association' should be revised. The polysome fractions are normal, and the data show that T492A only increases its presence at the ribosome and affects translation fidelity, while translation itself continues. Please revise the title to accurately reflect these findings.

19. Line 403: change "Notably, T492 occurs rapidly" to "Notably, T492 phosphorylation occurs rapidly."

20. Line 448: replace "Sis" with "Sis1."

21. Line 747: revise the sentence "Samples were diluted 6x with 50 mM tris pH 8.5 to reach <" to ensure it is grammatically correct.

22. Line 754: change "~1 ug" to "~1 g."

23. Line 756: define "i.d." in the context of the PepMap C18 particles.
24. Line 483: change "The yeast cell integrity pathway" to "The yeast cell wall integrity pathway."
25. Line 860: capitalize "V" in "vocalno" to "Volcano."
26. Line 863-864: Add a verb to the sentence "T492 phosphorylation does not the binding of intrinsically-disordered proteins."
27. Add the hour unit to the time of the AZC treatment in Figure 1f.
28. Revise "Translational pausing" to "Translational fidelity" in Figure 8.

Reviewer #4

(Remarks to the Author)

Reviewer #5

(Remarks to the Author)

In this interesting multidisciplinary study, Truman and colleagues make important contributions to our understanding of how distinct features of the budding yeast heat shock response, a longstanding model for the conserved eukaryotic HSR, are integrated. Among the notable observations is that phosphorylation of Ssa1 T492 is required for full Hsf1 activation (however, see below). If confirmed, this process, operating alongside the classic chaperone titration mechanism, can provide an explanation for the rapidity of Hsf1 activation which is detected within seconds – and as pointed out by the authors, too rapid to be explained by chaperone titration mediated by unfolded nucleoplasmic/cytosolic proteins alone. Additional strengths of the study are the multiple approaches used, the very high quality of data and the thoughtful writing (although see below).

My enthusiasm is tempered by several concerns that I believe need to be addressed before the manuscript is suitable for publication in Nat. Commun.

1. A key claim is that Hsf1 transcriptional activation is functionally linked to the phosphorylation status of Ssa1 T492. This is based on experiments conducted using a specialized strain in which the genes encoding all 4 cytosolic Hsp70's – SSA1, SSA2, SSA3, SSA4 – have been deleted and WT and mutant Ssa1 are expressed from multicopy centromeric plasmids. My concern is that the authors use these strains to assay Hsf1 activity indirectly, using either a luciferase reporter (Fig. 4b) or Western blot of a representative Hsf1 target gene (Fig. 4c). As later shown (Fig. 5d, e), the T492A mutation in Ssa1 strongly inhibits global translation (although unaddressed is the possibility that translation of Hsf1 target mRNAs is spared). Thus, the assays used for Hsf1 activation are potentially confounded by suppressed translation of HSP mRNAs (including the HSE-driven mRNA encoding the luciferase reporter) elicited by the same point mutation that obviates release of Hsf1 +HS (a striking observation shown in Fig. 4d). The authors need to use a more direct assay of Hsf1 activation (such as RT-qPCR or Pol II ChIP) to substantiate this key point.
2. As alluded above, the authors surmise that kinetics of phosphorylation of T492 are sufficiently rapid to explain the rapidity with which Hsf1 itself is activated in heat shocked yeast, a phenomenon that occurs within ~1 min (see, for example, PMID 28970326), too rapid to be explained by the classic chaperone titration model. If this is in fact the mechanism, then to prove it the authors need to fine-tune their kinetics analysis beyond what is presented in Fig 1d to include pertinent early time points.
3. A potential limitation is that Ssa1 T492A is expressed at a substantially higher level than WT Ssa1 (this needs to be quantified) in the experiments that employ the MH272 strain, which appear to be the vast majority. At the very least, the authors need to acknowledge this potential confounding aspect of the study.
4. A paradox is that interaction of Sis1 with Ssa1 is not affected by the T492A mutation, yet Sis1 is required to maintain Hsf1 in a transcriptionally inactive state (PMID 33326013). The authors need to address this.
5. The Introduction and (especially) Discussion need to be tightened up. At places, these sections read like review article rather than a research paper.

Minor points

1. The key study identifying Hsp70 as the principal chaperone that represses yeast Hsf1 is ref. 9. This, and a companion study (PMID 29393852), should be clearly acknowledged throughout the ms. (e.g., on lines 68, 279, 392). Likewise, the initial study identifying Hsp70 as a principal repressor of mammalian HSF1 is ref. 59. It too needs to be more clearly acknowledged.
2. Misinterpretation of data: lines 157-158 ("not detected" is inconsistent with data presented in Figs. 1c, d, e; 2e).
3. Criteria for identifying the presence of foci in Fig. 6 needs to be clearly stated. Pab1-GFP foci are visible in T492A cells under recovery conditions, yet this is not evident from the bar graph.
4. Figures 1a and 5c could be clearer through labelling of each of the three Hsp70 domains and the free, monosome and polysome peaks.
5. The authors are inconsistent in their reference to the role played by Msn2 in the yeast transcriptional response to stress. In places, they lump Msn2 with Hsf1 as intrinsic to the HSR; in other places they characterize Msn2 as driving a separate environmental stress response. Just be consistent.
6. Writing is for the part clear; however, there are multiple places where either punctuation or words are missing (and/or word choice is poor/misleading). For example: lines 99, 105, 141, 143, 154, 179, 190, 258, 382, 403, 471, 504, 864, 870, 877, 879 (antagonizes not "mediates"), 920, 949-950.

Reviewer #6

(Remarks to the Author)

Version 2:

Reviewer comments:

Reviewer #1

(Remarks to the Author)

Although the authors addressed many of the points raised by this reviewer, there are still some of them that were not sufficiently or adequately approached:

Point 4: Regarding the mechanism by which phosphorylation of Hsp70 by Pkc1 would trigger Hsf1-Hsp70 dissociation, and in response to my criticism about the absence of differences in the amount of Ssa1 or Ssa1- T492A co-immunoprecipitated with Hsf1, this could be due to the high amount of Ssa1 T492A present and not to differences in the affinity between Hsf1 and Ssa1. The authors used WT and T492A strains with plasmids under constitutive GPD promoters. It is true that, as shown from the input, in these strains, Ssa1 levels are equal. The authors claim in the rebuttal letter, "We repeated the Hsf1-Ssa1 interaction experiment using the new strains, and we see an even clearer T492-driven Hsf1 interaction difference. This data can be found as Fig. S3e". However, the data in Fig. S3e show: 1) If the amounts of immunoprecipitates loaded are equal in the WT and T492A strains, the amount of Ssa1 co-immunoprecipitated with Hsf1 is greater in the T492A than in the WT; and 2) Although a significant amount of Ssa1 remains bound to Hsf1 in the T492A strain at 39°C, there is a decrease in the amount of Ssa1 co-immunoprecipitated with Hsf1 at this temperature. Therefore, phosphorylation of T492 does not fully account for the dissociation. This needs to be explained and nuanced.

Point 8: Regarding the relationship between heat-induced phosphorylation of Hsp70 and the CWI pathway, I still believe that the conclusion about the role of Bck1 in the requirement for 408 Ssa1 T492 phosphorylation in promoting CWI signaling is overstated based on the results shown. The experiments should be conducted with the full-length Bck1. Surprisingly, the authors insist in the new version of the manuscript that no one has been able to express full-length Bck1, which is not true. In the rebuttal letter, they explain that they tried again to express the full-length Bck1 fused to Myc without success, and that Hruby et al experienced the same issue and also had to break Bck1 into two fragments. Nothing could be further from the truth. In the work by Hruby et al. (2011), the authors constructed several versions of the protein to analyze the role of the N-terminal regulatory domain, the C-terminal kinase domain, and residues 802-824 in various interactions, but they also expressed and analyzed a full-length C-terminally Myc-tagged version of Bck1-Myc (Supplementary Figure S1). Furthermore, they show that cooperation between the regulatory domain and the kinase region works together to achieve strong binding to Mkk1-Slt2, supporting the idea that full-length Bck1 is necessary to support these conclusions. This confirms the possibility of conducting these experiments. If the authors can not express full-length Bck1, they can request the strains that have already been published.

Reviewer #2

(Remarks to the Author)

Reviewer #3

(Remarks to the Author)

The authors have done a solid job addressing the many points raised by all reviewers. Concerns expressed by the majority of the reviewers were remarkably similar and consistent and the responses, both written and experimental, satisfyingly addressed the major issues. Importantly, several claims were toned down to more appropriately reflect the data. Overall, I am satisfied with the resubmitted version of the manuscript and feel that it will be an important addition to the stress, signaling literature.

Reviewer #4

(Remarks to the Author)

Reviewer #5

(Remarks to the Author)

The manuscript has been strengthened by inclusion of early time points in the Ssa1 phosphorylation analysis (lovely western blot in new Fig. S1) that provides compelling support for the authors' argument that Hsf1 activation is too rapid to be

explained by chaperone titration alone. Also strengthening the paper are experiments that express Ssa1 (WT) and Ssa1(T492A) under control of the GPD promoter that result in equivalent Ssa1 expression levels (Fig. S3a,b,c,e), and inclusion of the RNA abundance measurement in Fig. 4c (note that for clarity, this graph should be relabeled as “Luciferase mRNA levels”).

The above notwithstanding, the new RNA measurements in Fig. S3d raise concern. It is very odd that the SSA1 RNA level is not increased in WT cells exposed to a 30 min 39°C heat shock given that multiple studies have observed a HS-induced increase in SSA1 RNA (e.g., nascent SSA1 RNA levels are induced ~20-fold in cells shifted from 30° to 39°C during the first 5 min of heat shock (PMID 30332327; Suppl. Table 1)). Equally puzzling is the >10-fold higher SSA1 mRNA levels in the mutant compared to the WT; these too show no increase following the 30 min HS. It is possible that this experiment was confounded by the fact that the SSA1(WT) and SSA1(T492A) genes were borne on multicopy CEN plasmids. (Were the WT and T492A genes regulated by the WT SSA1 promoter and if so, how much upstream region was present? The designation “SSA” is unclear.) The authors’ explanation that the enhanced SSA1 RNA expression reflects compensatory activation of Msn2/Msn4 (lines 322-329; 442-443) is inconsistent with experiments that show SSA1 is only weakly transcribed (5% of WT levels) in heat shocked cells in which nuclear Hsf1 has been depleted using anchor away (PMID 30332327; Suppl. Table 1). While measurement of luciferase mRNA levels arising from the integrated HSE-Luciferase reporter gene (Fig. 4c) are consistent with Hsf1 activity being inhibited in cells expressing Hsp70(T492A), RNA measurements of Hsf1-dependent genes located at their native chromosomal locus would be extremely helpful to clarify this important point (namely, that Hsf1 bound by Ssa1(T492A) is incapable of driving transcriptional activation of its natural targets). Candidates for such an analysis include SSA2, HSC82, HSP82 and BTN2 as these genes are strongly Hsf1-dependent with little or no Msn2/4 contribution (Solis et al, Mol. Cell 2016; Pincus et al, MBoC 2018). Clinching the idea would be an Hsf1 ChIP analysis that demonstrates equal occupancy of Hsf1 at one or more of these genes in WT and Ssa1(T492A) cells following an acute HS. Additional points:

1. Was NHS temperature in the experiments described above set at 25°C? This important detail was never made clear in any of the figures. It needs to be.
2. Minor
 - a. Lines 1144-1145. Solis et al is incorrectly cited as Mol. Cell 2018. The year, volume and page numbers are incorrect.
 - b. Details of the RT-qPCR assay are more appropriate for the M&Ms, not figure legends.
 - c. Line 201: “arginine at the -3 position” is inconsistent with the sequences presented in Fig. 1b.
 - d. Inaccurate wording, typos, punctuation issues still exist throughout the ms. For example: lines 130, 142, 190, 310, 905, 913, 915, 920.

Reviewer #6

(Remarks to the Author)

We offer below our point-by-point responses to queries and comments on our submitted manuscript titled, "Mechanosensor-mediated Hsp70 phosphorylation orchestrates the landscape of the heat shock response". We would like to thank the reviewers for your constructive comments and very positive feedback. We believe that the revised manuscript and associated new data provide an improved and compelling story for publication in *Nature Communications*.

Response to Reviewer #1

We would like to thank the reviewer and trainee for their thoughtful review of our manuscript. We have performed additional experiments that clarify your queries and in doing so we believe the manuscript has been significantly improved.

1) The authors show that T492 phosphorylation of Ssa1 by heat shock depends on Mid2 but not Wsc1. Previous reports on the CWI sensors required for Mpk1 activation under heat shock conditions showed that both proteins (Mid2 and Wsc1) sense heat shock (Verna et al, PNAS, 1997; Gray et al, EMBO J, 1997; Ketela et al., J. Bacteriology, 1999). How do explain the authors these differences?

Despite extensive research, the cellular rationale for the presence of multiple upstream receptors in the CWI pathway remains unresolved. Nevertheless, the studies cited by the reviewer clearly demonstrate that deletion of *MID2* results in temperature-sensitive phenotypes and impaired Mpk1 phosphorylation. In this context, the *mid2Δ* strain phenocopies the *ssa1-T492A* mutant, underscoring the functional relevance of this signaling axis. While deletion of *WSC1* also confers temperature sensitivity, this observation is not contradictory but instead supports the notion of parallel or partially redundant signaling branches within the pathway. We have revised the Discussion section to clarify and incorporate these interpretations.

2) Regarding the experiments showing that Pkc1 mediates phosphorylation of Ssa1 at T492:
a) Figure S1 shows the cells' phenotype as validation of the Pkc1-AID strain, but levels of Pkc1 parallel to Phospho T492 (Figure 2d) need to be shown.
b) In the in vitro kinase assay (Figure 2f), Pkc1 was purified from cells growing at 25°C, but the effect of temperature (39°C) also needed to be assayed.

To address point a) We have now included a figure showing the levels of Pkc1-AID as requested using an AID antibody. This figure can be seen as Fig.S2b. For point b) the experiment was performed using Pkc1 purified from heat-treated cells (as is the standard protocol in the field). Our apologies for this error; we have now revised the text.

3) Regarding the Mass spec experiments comparing heat-shocked WT and T492A cells, I have several questions and comments (see below), but my main criticism is that it does not contribute much to the characterization of the functional role of Ssa1 T492 phosphorylation.

a) Although the objective is to compare and quantify differences in protein interactors of both proteins, using controls of untagged strains should avoid non-specific interactions.

We appreciate the reviewers comment about untagged strains. Over time, proteomic technologies, our experimental methodologies and our analysis methods have improved to the degree that we don't typically need to use an untagged control anymore. This has been our standard practice for all our publications on chaperone interactions from *PLOS Genetics* to *Cell*.

b) Details about how quantification was measured are missing: How many replicates were analyzed? What kind of normalization was applied?

We have addressed this in the manuscript.

Moreover, no experimental validation of any of the proteins identified was carried out. This is absolutely necessary, particularly in relation to some of the proteins whose functional relationship with Ssa1 phosphorylation is discussed throughout the work.

Proteomic technologies have advanced substantially over time to the degree that they are much more accurate than older methods like Western Blots. This together with the large number of interactions observed make it impractical to validate interactions using other methods. Some of the substantially changed interactions will be the focus of our group's future studies, but we believe this is beyond the scope of the current study.

c) In 784: fragment mass tolerance (0.2 Da) is too high (an adequate setting should be 0.02).

Yes, this is a typographical error, and we have now corrected it in the revised manuscript.

d) 647 proteins are identified and 560 quantified, but only 380 are included in the Excel file. Why? There was a text error; we have revised the manuscript.

e) The text through lines 256-262 has several errors: the mutation does not impact the association of fifteen chaperones and co-chaperones, but only nine. The other six do not show differences.

We have now addressed the errors and revised the manuscript.

f) One of the conclusions of the proteomic approach is that there are differences in the sets of client proteins bound to Ssa1 WT and T492A regarding the IDR composition of interactors. However, the authors do not clarify much about this conclusion.

We appreciate that the data were not clear for the IDPs. We have revised our analysis and show that the majority of Ssa1-IDP interactions are actually T492 phosphorylation state-independent.

4) From the results of experiments described in Figure 4d, the authors claim that T492 phosphorylation is important for Ssa1-Hsf1 dissociation upon heat shock. This is a critical point in explaining the mechanism by which phosphorylation of Hsp70 by Pkc1 would trigger Hsf1-Hsp70 dissociation, leading to the activation of the HSF response. However, I do not think the results

properly support this conclusion derived from the fact that mutation of T492 leads to the overexpression of Ssa1. Apparently, there are no differences in the amount of Ssa1 or Ssa1-T492A co-immunoprecipitated with Hsf1, but we can not exclude that this is due to the high amount of Ssa1 T492A and not to differences in the affinity between Hsf1 and Ssa1 as a consequence of the mutation. Experiments to measure differences in the affinity of Ssa1 variants for Hsf1 are needed. Additionally, reducing the amount of extract in the Co-IP and co-IP experiments in the opposite direction (immunoprecipitation with anti-Flag) could help. Moreover, Co-IP experiments should include a control of cells expressing only GFP to exclude possible nonspecific binding. Additionally, experiments showing that mutation of Thr 492 to Glu or Asp reverses the loss of dissociation should be appropriate.

In this study, we used native SSA promoters on our constructs to keep the conditions as physiologically relevant as possible. As the reviewer noted, we observe higher Ssa1 expression in the T492A expression compared to WT, which we believed to be due to overcompensation by Msn2/4 signaling. To address issues potentially caused by this overexpression, we reconstructed the WT and T492A strain using plasmids with constitutive GPD promoters. In these strains, Ssa1 levels are equal, confirming that the expression change previously seen was promoter-driven. Encouragingly, the GPD-driven Ssa1 strains display the same phenotypes as SSA-driven ones (see Fig. S3b). We repeated the Hsf1-Ssa1 interaction experiment using the new strains and we see an even clearer T492-driven Hsf1 interaction difference. This data can be found as Fig. S3e. The suggested experiment using the phosphomimic is challenging due to the inviability of the T392E mutant and there is no guarantee that T492E would mimic a truly phosphorylated state.

Contrary to Figure 4d's conclusion, the MS interactors results provided in Table S3 show that Hsf1 did not display a preference for Ssa1 T492A ($\text{Log}_2(\text{WT}/\text{T492A})=-0.075$), but this is not discussed.

It should be noted that Hsf1 has a very low cellular abundance (less than 2000 molecules per cell) and is typically hard to quantify by MS. We believe our revised experiments provide enough data to confidently say that T492 alters binding with Hsf1.

5) The authors argue (ln 300-306) that overexpression of Ssa1 T492A is due to an activation of a Msn2/4 transcriptional response, but from my point of view, this is an overestimated conclusion. The authors show that levels of Hsp12 are elevated in these circumstances, but no additional proof (i.e mRNA levels of SSA1 in a msn2/4 mutant strain; activation of Msn2/4; nuclear translocation) is shown. Moreover, the authors claim that their data demonstrate that T492 phosphorylation is important for the activation of both Hsf1 and Msn2/4-mediated heat shock-induced transcriptional response. However, levels of Hsp12 in the T492A strain are induced at 25°C, suggesting that Msn2/4 activation in this strain, if it exists, is not mediated by heat shock but induced constitutively.

As noted above, when we replace the promoter of our SSA promoter-driven Ssa1 construct with a constitutive GPD promoter, the levels of Ssa1 T492A are expressed at equivalent levels to WT.

This suggests the effect seen is transcriptional rather than degradation of the protein. In our T492A mutant, Hsp12 levels are constitutively high suggesting overactivation of the Msn2/4 pathway. The reviewer is correct that this response is not impacted by heat and this is because in the T492A mutant, heat-induced T492 phosphorylation cannot occur. We attempted to delete Msn2/4 in our T492A mutant background, but could not generate the strain. This is probably because loss of Hsf1 function in an Msn2/4 background is lethal (Mühlhofer et al., 2024). We have instead revised the text to take into account the reviewer's point.

It is also important to mention here that, in contrast to these results, previous works (Beltrao et al, Cell. 2012 Jul 20;150(2):413-25) did not report overexpression of Ssa1 in a T492A S495A strain. This is note even commented in the manuscript.

It should be noted that in the excellent Beltrao study all constructs are expressed via constitutive GPD promoters, whereas we used the SSA promoter to attempt to provide more native conditions. However as mentioned above, we have reconstructed the WT and T492A strains using the GPD promoter. In this strain, T492A has the same abundance as WT (while displaying all the same phenotypes). We have added this data to the revised manuscript and figures (Fig.S3a-b).

6) Regarding the results relating Hsp70 phosphorylation and ribosome association:

a) Conclusions from quantifying data from the Western blots in Figure 5d require further explanations about the number of replicates and statistical analysis of the differences. Otherwise, conclusions should be overestimated and not supported by the data.

We agree and have made the requested changes throughout.

b) As stated by the authors (In 310), Hsp70 dissociation from the ribosomes during heat shock promotes translational pausing. However, the author's results do not fit with this argument since they observe that, while polysome-bound Ssa1-T492A increases with respect to the WT, the T492A strain displays lower translation rates than the WT. Moreover, other publications studying the effect of phosphorylation of this residue report opposite conclusions. Moss et al. (Cell Host Microbe, 2019) reported that phosphorylation of the human Hsp70 in the equivalent residue (T495) increases its binding to polysomes, leading to translation inhibition. Additionally, Beltrao et al (2012), showed that the mutant T492A S495A was defective in polysome binding.

We appreciate the reviewer's careful reading and the opportunity to clarify the apparent discrepancy. Our original sentence implied that complete dissociation of Hsp70 from ribosomes is necessary for every instance of translational pausing. Recent work as well as our own data indicate a more nuanced model in which either excessive association or complete dissociation of cytosolic Hsp70 can slow translation, depending on the chaperone's nucleotide state and the stage of elongation. We have therefore revised the text to read: "Heat shock remodels Hsp70-ribosome interactions, and both transient dissociation and persistent over-association of Hsp70 have been reported to impede elongation, thereby contributing to global translational pausing

(Shalgi et al., 2013).” In addition, our data (Fig. 5d-f) show that the unphosphorylatable T492A variant remains locked on heavy polysomes during heat shock. This persistent binding is predicted to hinder elongation factor access, slow ribosome translocation and reduce methionine-analog (HPG) incorporation, exactly as we observe. Importantly, polysome accumulation in the mutant is accompanied by slower ribosome run-off in time-course profiles (new Supplementary Fig. S4c), indicating stalled, not active, ribosomes. Thus, higher polysome-bound Ssa1 does not equal higher productive translation.

Our results differ slightly from Moss et al. 2019 and Beltrao et al (2012 probably due to the systems and experimental conditions tested. For example, Beltrao’s double (T492A/S495A) or triple mutants severely altered the substrate-binding groove geometry, likely compromising ribosome affinity altogether. In contrast, our single-site mutant preserves the binding interface but blocks the ATP-regulated release step, producing hyper-association. Both our study and Moss et al. examined chaperone behaviour under acute stress (heat or LegK4 effector), conditions under which stalled ribosomes accumulate. Beltrao et al. assayed cells in steady-state growth, where Hsp70 turnover on ribosomes is faster and phosphorylation may promote initial recruitment rather than release. Finally, Human Hsc70 lacks the yeast-specific co-chaperone RAC-Ssb system; nevertheless, the convergence between Moss et al. and our yeast data-*increased* polysome binding when T492/T495 cannot be dephosphorylated supports a conserved “release” function for this modification.

c) Looking at the graphics in Figure 5f, it seems that the threefold increase in the mutant referenced in the text (ln 339) is overestimated. Additionally, at what temperature were these experiments performed? Is this effect a consequence of heat shock?

We have resolved this in the text.

d) In the legend of Figure 5f, they mention that statistical significance was calculated via ANOVA. However, they only have two groups, so this is not an appropriate analysis.

We apologize. We have resolved this in the text.

7) Regarding the experiments of P-body formation upon heat shock and disassembly:

a) The description of these results is not acceptable. What was the heat shock temperature used in these experiments?. Looking at the legend of Figure 6, the experiments were carried out at 46°C for 15 min and 30 min. However, looking at the methods section they say that these experiments were carried out at 39°C. Moreover, heat shock recovery in Figure 6 is indicated at 30°C during 60 and 120 min, but in methods, they describe 1 or 2 hours of heat shock at 39°C. Stress granules in response to heat shock (46°C) have been described in several reports but at shorter times (10 min).

We apologize, this was a labeling error. We have corrected the labels for these figures.

a) The effect of the T492A mutation in Pab1 foci disassembly shown in the graphics of Figure 6 does not correspond to the pictures: in the picture of Pab1/WT Ssa1 in recovery (30°C) there are almost no foci.

b) How do the authors explain that in the majority of the pictures stress granules (Pab1) co-localize with P-bodies (Edc3)?

We agree that these proteins can be seen to colocalize and this reflects well-documented biology rather than a technical issue. P-Bodies are known to nucleate early and can seed stress granule assembly (Kedersha et al.2005 and Buchan et al. 2008).

8) Regarding the relationship between heat-induced phosphorylation of Hsp70 and CWI pathway, the authors show that heat shock-mediated phosphorylation of Hsp70 in T492 via Pkc1 is absolutely necessary for the activation of Mpk1 and for the transcriptional response via the CWI pathway. This is quite interesting and innovative. However, the experiments developed to find out the mechanism responsible for this connection do not support the conclusions, and the mechanism is not characterized:

a) The authors focus on the possibility that the effect of Ssa1 T492A mutation on Mpk1 activation can be a consequence of differences in the levels of the protein kinases Mkk1, Mkk12, and Bck1, but there are many other possibilities.

Our data strikingly show that Mpk1 phosphorylation is lost in T492A mutants. Given the role of chaperones in stabilizing client proteins, it is a logical assumption that a CWI protein is a client of Ssa1. Mkk1/2 abundance are not impacted in T492A, so we logically went upstream to Bck1. Our data clearly show that Ssa1 interacts with Bck1 and that Bck1 domains are destabilized in T492A.

b) Even focusing only on this possibility, the results do not support the conclusions:

- They work with two fragments of Bck1 instead of the full-length protein because they were unsuccessful in detecting Bck1 even in WT cells (In 373). Even they argue explicitly (In 375) that since Bck1 discovery, there are no reports containing Western blots of this protein. However, this is not true: to my knowledge at least in two publications full-length versions of Bck1 fused to Myc (Hruby et al, J Cell Sci, 2011, 124 (1):35-46; García et al, J Cell Sci,2016, 129: 1649-1660) have been reported, so all the experiments using the Bck1 protein fragmented in two pieces (Figure 7 g, h, i) are not justified.

Bck1 is an extremely challenging protein to work with. It is rather large, roughly 165KDa but with a corresponding low abundance of around 1200 molecules per cell. In Hruby et al, the authors also experienced the same issue as us and had to break Bck1 into fragments (1-800 aa, see Figure 2 of that paper). In Garcia et al, the authors claim that they natively tagged Bck1 and Pkc1 with a Myc tag. However if the reviewers look at Figure 3, Bck1 and Pkc1 are detected at the same MW-unusual given that these proteins have distinctly different molecular weights. In any case, we tried our best to study Bck1 as a full-length protein. As mentioned in the text, we attempted to tag it on the genome as in Garcia et al and did not detect the protein. We turned

next to the ZZ-tagged collection and did not see any protein even under the galactose-driven promoter. We even fully sequenced and validated this plasmid to check that there were no frame shifts present. Fortunately, we were successful in detecting Bck1 when expressing domains as in Hruby et al. We have adjusted the text in the manuscript to reflect the limitations of our results.

•Moreover, the authors describe in line 238 that co-IP experiments were done in WT and T492A strains but no CoIP of the mutant is shown in Figure 7.

Resolved in the revised manuscript.

•In any case, the decrease in the levels reported in for the carboxy fragment can not account for the complete blocking of Mpk1 activation and lack of induction of FKS2 and PRM5.

It is important to remember that for us to detect Bck1, we have substantially overexpressed the protein compared to native levels. While we agree this is not ideal, it was necessary for this experiment. Our working model is that the impact of T492A would be more substantial on the native protein. We have edited the manuscript to account for this.

9) There are many errors and an absence of information in the majority of the figure legends
-Figure 1a: The predicted structure of Ssa1 is shown, but no reference is provided for how or where this prediction was obtained.

This has been resolved in the manuscript.

-Figure 1d: The Western blot only shows WT cells, despite the figure legend mentioning both WT and T492A cells.

This has been resolved in the manuscript.

-Figure 1e: The legend states that heat shock was performed at the indicated time points, but it was only performed at a single time point (1 h), as mentioned in the Results section.

This has been resolved in the manuscript.

-Figure 2d: The legend states that cells were treated with auxin and incubated at 39°C for different times. However, the actual experiment involved auxin treatment for the indicated times, followed by a 30-minute incubation at 39°C for all samples.

This has been resolved in the manuscript.

-Figure 2f: the legend mentions the use of antisera to FLAG epitope, but the epitope shown in the figure is HIS.

This has been resolved in the manuscript.

-Figure 3e: In line 864, a verb is missing, making the conclusion unclear.

This has been resolved in the manuscript.

-Figure 4d: The legend does not mention the antisera used for the PGK1 epitope.

This has been resolved in the manuscript.

-Figure 4f: The legend mentions the use of antisera to PGK1 epitope, but the epitope shown in the figure is GAPDH.

This has been resolved in the manuscript.

-Figure 5f: The temperature at which the experiment was performed is not indicated.

This has been resolved in the manuscript.

Additionally, the legend mentions an ANOVA analysis, which is not appropriate since only two groups are being compared.

This has been resolved in the manuscript.

-Figure 7i: The legend states that "the C-terminus of Bck1 interacts with Bck1," but it should be "the C-terminus of Bck1 interacts with Ssa1."

This has been resolved in the manuscript.

-Figure S1: It needs further explanation

This has been addressed in the revision.

10) Regarding the "Materials and Methods" section, I found several inconsistencies and a lack of information necessary for experimental reproducibility.

-It is stated that reagents are provided in the supplementary tables, but this table is missing.

-The description of plasmid construction for the expression of Flag-Ssa2, Flag-Ssa3, Flag-Ssa4, and the HA-tagged Bck1 fragments lacks necessary details.

These plasmids were constructed by Genscript via gene synthesis. We have now included the full sequences of all the plasmids generated in this study in the supplemental data.

-The luciferase assay is described twice in this section, with inconsistencies between the two versions.

This has been resolved in the manuscript.

- In the first description (line 579), cells were grown at 25°C, while in the second (line 648), they were grown at 30°C.

This has been resolved in the manuscript.

- The first version states that incubation lasted 90 minutes with measurements taken every 5 minutes, whereas the second version states that incubation lasted 90 minutes with readings every 3 minutes.

This has been resolved in the manuscript.

- Both descriptions mention incubation at 37°C, but the correct temperature is 39°C according to the graph.

This has been resolved in the manuscript.

-In vitro kinase assay: The composition of the kinase buffer and loading buffer is not specified.

This has been resolved in the manuscript.

-Pab1 and Edc1 foci quantification. According to the figure, cells were heat-shocked at 46°C for 15 and 30 minutes, followed by recovery at 30°C for 1 and 2 hours. However, the Materials and Methods section states that cells were incubated at 39°C for 15, 30, 60, or 120 minutes, which does not match the figure.

This has been resolved in the manuscript.

Reviewer #3 (Remarks to the Author)

This manuscript investigates the role of threonine 492 phosphorylation of yeast Hsp70 (Ssa1) during heat stress. The authors demonstrate that heat shock induces Ssa1 T492 phosphorylation, dependent on the cell wall sensor Mid2 and the Pkc1 kinase of the cell wall integrity (CWI) pathway. This phosphorylation is shown to be essential for Hsf1 release and proper activation of the heat shock response (HSR), potentially contributing to its rapid kinetics. Furthermore, T492 phosphorylation influences Ssa1's ribosome association and translation fidelity during heat stress, and regulates P-body disassembly during recovery. Notably, the CWI pathway is not activated in T492A mutant cells during heat shock, though a direct mechanistic link remains to be established.

This study demonstrates a novel and extremely intriguing connection between cell-wall integrity pathway-mediated Hsp70 phosphorylation and regulation of the heat shock response. It elucidates how environmental stress signals are transmitted from cell surface sensors to the nucleus, activating the stress response, thereby contributing to our understanding of the chaperone code. Overall, much of the data are very compelling and the quality of the results is high. However, open questions remain, and some of the results require additional support, as detailed below.

1. Please specify the heat shock duration for all the related Figures or at least mention it in the figure legends.

This has been resolved in the manuscript.

2. Figures 1c, 1d, 1e and 2e indicate a low level of T492 phosphorylation without heat shock. This should be explicitly recognized in the text. The authors should also discuss potential kinases and the functional implications of this basal phosphorylation.

This basal level of Ssa1 phosphorylation is to be expected given that it is almost impossible to have a completely “unstressed” cell. The cell integrity pathway is always active at low levels even at 25°C, which would explain the almost undetectable basal levels of T492 phosphorylation.

3. In Figure 2b, residual T492 phosphorylation is observed in the absence of Mid2. To ensure reproducibility of this finding, the authors should repeat this experiment a minimum of three independent times and present the average signal, similar to that in Figures 4c and 4f, or clarify if this was indeed done. Additionally, given the potential for delayed responses, this experiment should be conducted using the 4-hour timeline employed in Figure 1c to definitively rule out any delayed phosphorylation kinetics. The authors should also discuss why in the absence of Mid2, Wsc1 could not activate Pkc1 to phosphorylate Ssa1. It is unclear what significance should be attached to the apparent Mid2 specificity, if any.

We have resolved this in the text.

4. In Figure 2d, the authors should examine Pkc1 protein levels upon auxin induction at the indicated time points to strengthen the correlation between T492 phosphorylation reduction and Pkc1 levels. This could be achieved by using anti AID tag antibody or tagging Pkc1 with additional small epitope tag, such as FLAG or HA. Furthermore, if generating a *pkc1* null mutant in sorbitol-containing media is feasible, utilizing this mutant directly would provide a more straightforward and definitive assessment of Pkc1's role in T492 phosphorylation.

We agree with the reviewer and have included a blot showing the level of Pkc1-AID (Fig. S2b). As expected, the degradation of Pkc1 tracks with loss of T492 phosphorylation.

5. While Figure 2f demonstrates that Pkc1 can phosphorylate Ssa1 in vitro, given that T492 is within the preferred PKC substrate motif, in vivo validation of this interaction is important. To

confirm this interaction within cells, a crosslinking and pulldown experiment using Pkc1 as bait to detect Ssa1 association and T492 phosphorylation should be considered. Notably, Pkc1 was absent from the Ssa1 interactors identified by mass spectrometry (MS) (Supplementary S1). This discrepancy requires discussion within the text.

We believe that our cellular experiments (Fig. 2d, e, f) demonstrate that Pkc1 function (but not Bck1, Mkk1/2 or Mpk1) is required for T492 phosphorylation. Given the additional evidence from the in vitro experiment we are confident that Pkc1 is the direct kinase for T492, especially given the substrate motif. It should be noted that kinase interactions are notoriously transient and that Pkc1 abundance in yeast is very low. Under these conditions, we would not expect to see Pkc1 as an interactor by MS.

6. The authors noted that Ssa1 T492A exhibits significantly higher expression (at least five-fold) compared to Ssa1 WT (Figures 1c, 2c, 4d). In Figure 3, where MS data and changes in Ssa1 interactors are presented, it is unclear whether the levels of detected interactors were normalized to the respective levels of Ssa1 WT and Ssa1 T492A. If normalization was not performed, the observed preferential binding of interactors requires re-evaluation to account for the differential expression of Ssa1 variants.

This is a very important point and we thank the reviewer for bringing this up. We have over a decade of experience in characterizing the changing interactions of chaperones in response to stress and mutations¹⁻⁸. In all of these studies (this one included), we normalized all the interactome data between the sets on the IP's protein, in this case, Ssa1. We have clarified this in the text.

7. The observed growth defect of the T492A mutant at 37 °C (Figure 4a) and the corresponding lack of Hsf1-mediated heat shock response (Figures 4b, c) are very intriguing. However, given the significantly higher expression of T492A compared to WT (at least five-fold), it is possible that the observed Hsf1 trapping is an artifact of T492A overexpression, rather than a direct effect of the T492A mutation itself. To address this, we recommend reducing T492A expression using a weaker heterologous promoter and examining whether Hsf1 still fails to be released under heat shock conditions.

We agree with the reviewer. We reconstructed the strains on a weaker but constitutive promoter (GPD). These new strains display the same phenotypes as before while being equalized for Ssa1 abundance (Fig.S3a-b). We repeated the Ssa1-Hsf1 IP with these new strains and observed an even clearer T492-mediated interaction change. This new data can be found as (Fig. S3e).

8. Additionally, regarding Figure 4A, the observed temperature sensitivity at 37°C appears to be significantly remediated by the addition of sorbitol, an osmotic stabilizer. This finding suggests that the growth defect is not due to a reduced heat shock response, as mentioned in the text, but mostly due to the clear defects in CWI signaling shown in Figure 7. This in some way alters the interpretation of the effects of T492 phosphorylation in both the results and discussion and should be clearly explained.

This is an excellent point and fortunately work we published over 20 years ago nicely explains this observation. Hsf1 controls the levels of several chaperones including Hsp90, a chaperone that stabilizes the Mpk1 kinase. Truncating Hsf1 (*hsf1-583* mutant) in yeast results in lowered Hsp90 levels. Mpk1 is a client of the Hsp90 system and we found that in *hsf1-583* Mpk1 is still dually phosphorylated but is unable to phosphorylate its downstream target Rlm1 (Truman et al. 2007). Because the terminal issue with the *hsf1-583* mutant is loss of Mpk1 function, the *ts* nature of this mutant can be suppressed on media containing sorbitol. Our data shows that T492 phosphorylation regulates multiple components of the heat shock response cell integrity signaling, providing a novel paradigm tying these pathways together.

9. In Figure 4d, the signal for WT Ssa1 appears relatively weak. To ensure the reliability of the observed reduction in Ssa1-Hsf1 binding upon heat shock, we recommend repeating this experiment at least two additional times and quantifying the average signal. Furthermore, overexpressing WT Ssa1 may enhance the signal and provide a more robust demonstration of the reduction in Hsf1 binding during heat shock.

We agree. We have the IP results and have provided new data with strains expressing Ssa1 at similar levels (see Fig. S3e).

10. The observation of Msn2/4 activation in the T492A mutant under normal conditions (Figure 4f) suggests a complete blockage of Hsf1 activity. This raises crucial questions regarding the mechanism of T492A's effect. Specifically, it is necessary to determine whether T492A expression is upregulated by the activated Msn2/4 pathway or the T492A mutation stabilizes the Ssa1 structure, thereby enhancing its apparent Hsf1 binding affinity.

We agree and as mentioned above have created new strains that express Ssa1 on constitutive GPD promoters. In these strains, WT and T492A Ssa1 are expressed equally. In addition we have completed RT-PCR analyses demonstrating that the transcription of Ssa1 is upregulated in the T492A strain (Fig.3c-d). Taken together, this confirms that the difference between WT and T492A Ssa1 abundance in the original strains was caused by transcriptional changes.

12. The observed inactivation of the CWI pathway in the T492A mutant (Figure 7a-c) is an interesting finding. However, the *in vivo* interaction of Ssa1 with Bck1 fragments (Figure 7h, i) raises a significant discrepancy, as Bck1 was not identified as an Ssa1 interactor in the mass spectrometry data (Supplementary table S1). The authors must address this discrepancy and clarify whether the Bck1 fragments represent the native state of Bck1 or potentially misfolded states that attract Ssa1 binding, taking into consideration the challenges inherent with working with the full-length protein. Furthermore, the phosphorylation status of Mkk1/Mkk2 in T492A cells under heat shock condition should be evaluated to further elucidate the CWI pathway inactivation.

The native level of Bck1 is extremely low (approx. 1000 molecules per cell compared to Ssa1) making it unlikely to be detectable by MS. We appreciate using domains of Bck1 are not ideal,

but this was the only way we can detect this challenging protein. We have revised the manuscript text to discuss the challenges and pitfalls of our working model.

Minor comments:

13. Introduce the full terms for AZC and CPZ upon their first mention in the text.

This has been resolved in the manuscript.

14. Add periods at the end of sentences on lines 99, 167, 467, and 716.

This has been resolved in the manuscript.

15. Use subscript for numerical values in chemical formulas (e.g., NaN_3 , N_2 , CuSO_4 , MgCl_2).

This has been resolved in the manuscript.

16. Common chemical names, such as 'glycerol,' should not be capitalized unless they appear at the beginning of a sentence.

This has been resolved in the manuscript.

17. Line 685: Remove the extra ")".

This has been resolved in the manuscript.

18. Line 308: The title 'Hsp70 phosphorylation is critical for protein translation and ribosome association' should be revised. The polysome fractions are normal, and the data show that T492A only increases its presence at the ribosome and affects translation fidelity, while translation itself continues. Please revise the title to accurately reflect these findings.

We agree that the original subsection title overstated the impact on bulk protein synthesis and did not highlight the fidelity phenotype we uncovered. We have therefore replaced the title to "T492 phosphorylation fine-tunes Ssa1 ribosome engagement and translation fidelity during heat shock".

19. Line 403: change "Notably, T492 occurs rapidly" to "Notably, T492 phosphorylation occurs rapidly."

This has been resolved in the manuscript.

20. Line 448: replace "Sis" with "Sis1."

This has been resolved in the manuscript.

21. Line 747: revise the sentence "Samples were diluted 6x with 50 mM tris pH 8.5 to reach <" to ensure it is grammatically correct.

This has been resolved in the manuscript.

22. Line 754: change "~1 ug" to "~1 g."

This has been resolved in the manuscript.

23. Line 756: define "i.d.." in the context of the PepMap C18 particles.

Details have been added.

24. Line 483: change "The yeast cell integrity pathway" to "The yeast cell wall integrity pathway."

This has been resolved in the manuscript.

25. Line 860: capitalize "V" in "vocalno" to "Volcano."

This has been resolved in the manuscript.

26. Line 863-864: Add a verb to the sentence "T492 phosphorylation does not the binding of intrinsically-disordered proteins."

This has been resolved in the manuscript.

27. Add the hour unit to the time of the AZC treatment in Figure 1f.

Completed

28. Revise "Translational pausing" to "Translational fidelity" in Figure 8.

Completed

Reviewer #5 (Remarks to the Author):

In this interesting multidisciplinary study, Truman and colleagues make important contributions to our understanding of how distinct features of the budding yeast heat shock response, a longstanding model for the conserved eukaryotic HSR, are integrated. Among the notable observations is that phosphorylation of Ssa1 T492 is required for full Hsf1 activation (however, see below). If confirmed, this process, operating alongside the classic chaperone titration mechanism, can provide an explanation for the rapidity of Hsf1 activation which is detected within

seconds – and as pointed out by the authors, too rapid to be explained by chaperone titration mediated by unfolded nucleoplasmic/cytosolic proteins alone. Additional strengths of the study are the multiple approaches used, the very high quality of data and the thoughtful writing (although see below).

My enthusiasm is tempered by several concerns that I believe need to be addressed before the manuscript is suitable for publication in Nat. Commun.

1. A key claim is that Hsf1 transcriptional activation is functionally linked to the phosphorylation status of Ssa1 T492. This is based on experiments conducted using a specialized strain in which the genes encoding all 4 cytosolic Hsp70's – SSA1, SSA2, SSA3, SSA4 – have been deleted and WT and mutant Ssa1 are expressed from multicopy centromeric plasmids. My concern is that the authors use these strains to assay Hsf1 activity indirectly, using either a luciferase reporter (Fig. 4b) or Western blot of a representative Hsf1 target gene (Fig. 4c). As later shown (Fig. 5d, e), the T492A mutation in Ssa1 strongly inhibits global translation (although unaddressed is the possibility that translation of Hsf1 target mRNAs is spared). Thus, the assays used for Hsf1 activation are potentially confounded by suppressed translation of HSP mRNAs (including the HSE-driven mRNA encoding the luciferase reporter) elicited by the same point mutation that obviates release of Hsf1 +HS (a striking observation shown in Fig. 4d). The authors need to use a more direct assay of Hsf1 activation (such as RT-qPCR or Pol II ChIP) to substantiate this key point.

We understand the reviewer's points that the altered luciferase activity seen in T492A may be a result of translational effects. We have completed the requested RT-PCR studies and these can be found in Fig. 4 and Fig. S3.

2. As alluded above, the authors surmise that kinetics of phosphorylation of T492 are sufficiently rapid to explain the rapidity with which Hsf1 itself is activated in heat shocked yeast, a phenomenon that occurs within ~1 min (see, for example, PMID 28970326), too rapid to be explained by the classic chaperone titration model. If this is in fact the mechanism, then to prove it the authors need to fine-tune their kinetics analysis beyond what is presented in Fig 1d to include pertinent early time points.

Despite the extreme technical challenges of this suggestion, we have now demonstrated that robust activation of this site occurs even 1 minute (!) of heat shock. This has now been added as Fig.S1.

3. A potential limitation is that Ssa1 T492A is expressed at a substantially higher level than WT Ssa1 (this needs to be quantified) in the experiments that employ the MH272 strain, which appear to be the vast majority. At the very least, the authors need to acknowledge this potential confounding aspect of the study.

In this study, we used native SSA promoters on our constructs to keep the conditions as physiologically relevant as possible. As the reviewer noted, we observed a clear increase in Ssa1-T492A expression compared to WT, which we believed to be due to overcompensation by Msn2/4

signaling. To address issues potentially caused by this overexpression, we reconstructed the WT and T492A strain using plasmids with constitutive GPD promoters. In these strains, Ssa1 levels are equal, confirming that the expression change previously seen was promoter-driven. Encouragingly, the GPD-driven Ssa1 strains display the same phenotypes as SSA-driven ones (see Fig. S3b). We repeated the Hsf1-Ssa1 interaction experiment using the new strains, and we see an even clearer T492-driven Hsf1 interaction difference. This data can be found in Fig. S3e.

4. A paradox is that interaction of Sis1 with Ssa1 is not affected by the T492A mutation, yet Sis1 is required to maintain Hsf1 in a transcriptionally inactive state (PMID 33326013). The authors need to address this.

We agree this is an important point, and our data resolve the apparent paradox by separating assembly of the Hsf1–Hsp70 repressor complex (which depends on Sis1) from the heat-induced dissociation step (which depends on Ssa1 T492 phosphorylation and nucleotide-exchange dynamics). Our epichaperome proteomics show that the T492A mutation diminishes Ydj1 binding but does not alter Sis1 association with Ssa1, indicating that Sis1-dependent recruitment/maintenance of the repressor complex remains intact in the mutant. Our Co-IP experiments demonstrate that, in WT cells, heat shock triggers Ssa1–Hsf1 dissociation, whereas in T492A cells this dissociation fails—pinpointing the block at the release step rather than at Sis1-mediated assembly. Finally, overexpression of nuclear Sse1 (Sse1-NLS), which accelerates Hsp70 substrate release in the nucleus, suppresses the temperature-sensitive phenotype of T492A, further supporting that T492 phosphorylation licenses efficient Hsf1 release rather than governing Sis1-dependent assembly. Overall, our results indicate that Pkc1-dependent phosphorylation of Ssa1 at T492 is required to trigger rapid release of Hsf1 on heat shock, a step that is mechanistically distinct from Sis1’s role. Consistent with this view, we note in the Discussion that T492-dependent remodeling preserves Sis1 interactions while altering others (e.g., Ydj1), reinforcing that the phosphorylation event fine-tunes co-chaperone usage without eliminating Sis1 function.

Minor points

1. The key study identifying Hsp70 as the principal chaperone that represses yeast Hsf1 is ref. 9. This, and a companion study (PMID 29393852), should be clearly acknowledged throughout the ms. (e.g., on lines 68, 279, 392). Likewise, the initial study identifying Hsp70 as a principal repressor of mammalian HSF1 is ref. 59. It too needs to be more clearly acknowledged.

We have added the appropriate references.

2. Misinterpretation of data: lines 157-158 (“not detected” is inconsistent with data presented in Figs. 1c, d, e; 2e).

We agree and have edited the manuscript accordingly.

3. The authors are inconsistent in their reference to the role played by Msn2 in the yeast transcriptional response to stress. In places, they lump Msn2 with Hsf1 as intrinsic to the HSR; in

other places they characterize Msn2 as driving a separate environmental stress response. Just be consistent.

We have adjusted the text to state that activation of Hsf1 and Msn2/4 are overlapping but distinct parts of the heat shock response.

4. Writing is for the most part clear; however, there are multiple places where either punctuation or words are missing (and/or word choice is poor/misleading). For example: lines 99, 105, 141, 143, 154, 179, 190, 258, 382, 403, 471, 504, 864, 870, 877, 879 (antagonizes not “mediates”), 920, 949-950.

Reviewer #6 (Remarks to the Author):

Thank you for taking time to review this manuscript, your thoughtful comments have helped us to improve the study.

References

1. Omkar, S. *et al.* Acetylation of the yeast Hsp40 chaperone protein Ydj1 fine-tunes proteostasis and translational fidelity. *PLoS Genet.* **20**, e1011338 (2024).
2. Knighton, L. E., Nitika, Wolfgeher, D., Reitzel, A. M. & Truman, A. W. Dataset of *Nematostella vectensis* Hsp70 isoform interactomes upon heat shock. *Data Brief* **27**, 104580 (2019).
3. Xu, L. *et al.* Rapid deacetylation of yeast Hsp70 mediates the cellular response to heat stress. *Sci. Rep.* **9**, 16260 (2019).
4. Truman, A. W. *et al.* Quantitative proteomics of the yeast Hsp70/Hsp90 interactomes during DNA damage reveal chaperone-dependent regulation of ribonucleotide reductase. *J. Proteomics* **112**, 285–300 (2015).
5. Truman, A. W. *et al.* CDK-dependent Hsp70 Phosphorylation controls G1 cyclin abundance and cell-cycle progression. *Cell* **151**, 1308–1318 (2012).
6. Dunn, D. M. *et al.* C-Abl mediated tyrosine phosphorylation of Aha1 activates its co-chaperone function in cancer cells. *Cell Rep.* **12**, 1006–1018 (2015).
7. Woodford, M. R. *et al.* Mps1 mediated phosphorylation of Hsp90 confers renal cell carcinoma sensitivity and selectivity to Hsp90 inhibitors. *Cell Rep.* **14**, 872–884 (2016).

Second Response to Reviewers

We offer below our point-by-point responses to reviewer queries and comments on our submitted manuscript titled, “Mechanosensor-mediated Hsp70 phosphorylation orchestrates the landscape of the heat shock response”. We would like to thank the reviewers for your constructive comments and positive feedback. We believe that this newly revised manuscript and additional clarifying data provide an improved and compelling story for publication in *Nature Communications*.

Response to Reviewer #1

1) Although a significant amount of Ssa1 remains bound to Hsf1 in the T492A strain at 39°C, there is a decrease in the amount of Ssa1 co-immunoprecipitated with Hsf1 at this temperature. Therefore, phosphorylation of T492 does not fully account for the dissociation. This needs to be explained and nuanced .

We agree and have edited the discussion appropriately.

2) The experiments should be conducted with the full-length Bck1.

We agree and have now included a blot of full-length Bck1 levels in WT and T492A cells. In agreement with our previous data, levels of Bck1 are decreased in T492A cells (see Fig. 6g).

Response to Reviewer #3

1) The authors have done a solid job addressing the many points raised by all reviewers. Concerns expressed by the majority of the reviewers were remarkably similar and consistent and the responses, both written and experimental, satisfyingly addressed the major issues. Importantly, several claims were toned down to more appropriately reflect the data. Overall, I am satisfied with the resubmitted version of the manuscript and feel that it will be an important addition to the stress, signaling literature.

We thank the reviewer for their kind comments.

Response to Reviewer #5

1) The manuscript has been strengthened by inclusion of early time points in the Ssa1 phosphorylation analysis (lovely western blot in new Fig. S1) that provides compelling support for the authors’ argument that Hsf1 activation is too rapid to be explained by chaperone titration alone. Also strengthening the paper are experiments that express

Ssa1(WT) and Ssa1(T492A) under control of the GPD promoter that result in equivalent Ssa1 expression levels (Fig. S3a,b,c,e), and inclusion of the RNA abundance measurement in Fig. 4c (note that for clarity, this graph should be relabeled as “Luciferase mRNA levels”).

We thank the reviewer for their kind comments. We have made the requested edits.

2) It is very odd that the SSA1 RNA level is not increased in WT cells exposed to a 30 min 39°C heat shock given that multiple studies have observed a HS-induced increase in SSA1 RNA (e.g., nascent SSA1 RNA levels are induced ~20-fold in cells shifted from 30° to 39°C during the first 5 min of heat shock (PMID 30332327; Suppl. Table 1)). Equally puzzling is the >10-fold higher SSA1 mRNA levels in the mutant compared to the WT; these too show no increase following the 30 min HS. It is possible that this experiment was confounded by the fact that the SSA1(WT) and SSA1(T492A) genes were borne on multicopy CEN plasmids. (Were the WT and T492A genes regulated by the WT SSA1 promoter and if so, how much upstream region was present? The designation “SSA” is unclear.)

Regarding the promoter, -554 to -1bp upstream of the gene was used and the full sequence of the plasmids are provided in the supplemental table. There are few factors that may account for the differences in the mentioned paper and our results. The strain backgrounds are very different-for example, our strain lacks Ssa2-4 and as the reviewer highlights the genes were borne on CEN plasmids. In addition, Our qPCR captures steady-state RNA at 30 min (not nascent RNA at 5 min) and therefore is not directly comparable to the early nascent-RNA measurements cited by the reviewer. It should be noted that our data are also consistent with our Western Blots and previously published literature where heat-induced changes in Ssa1 expression are not observed.

3) The authors' explanation that the enhanced SSA1 RNA expression reflects compensatory activation of Msn2/Msn4 (lines 322-329; 442-443) is inconsistent with experiments that show SSA1 is only weakly transcribed (5% of WT levels) in heat shocked cells in which nuclear Hsf1 has been depleted using anchor away (PMID 30332327; Suppl. Table 1).

When both alleles are expressed from the constitutive GPD promoter, SSA1 transcript levels are equal in WT and T492A (no heat-induction expected), showing that the large difference arises upstream of coding sequence at the native SSA promoter and/or post-transcriptionally (Fig. S3c). The anchor away data the reviewer cites is hard to directly compare to ours because of the differences in the strains. In the anchor-away system, the addition of rapamycin triggers the export of Hsf1 from the nucleus. In our mutant

T492A becomes locked onto Hsf1. This difference may be the source of altered SSA1 transcription.

4) RNA measurements of Hsf1-dependent genes located at their native chromosomal locus would be extremely helpful to clarify this important point (namely, that Hsf1 bound by Ssa1(T492A) is incapable of driving transcriptional activation of its natural targets). Clinching the idea would be an Hsf1 ChIP analysis that demonstrates equal occupancy of Hsf1 at one or more of these genes in WT and Ssa1(T492A) cells following an acute HS.

We thank the reviewer for this suggestion. We have included additional RNA measurements of BTN2 and HSP104 in WT and T492A cells (Fig. 4c). In agreement with our previous data (Fig. 4c), expression of these Hsf-mediated genes are decreased in the T492A mutant. Due to constraints on time, expertise and resources, we are unable to carry out the suggested CHIP experiment. However, we are confident that our new and previously submitted data clearly demonstrate that the T492A mutant displays a defect in Hsf1 activity.

5) Was NHS temperature in the experiments described above set at 25°C? This important detail was never made clear in any of the figures.

We apologize for this omission, yes, NHS was 25°C. We have updated the methods section to clarify this.

6) Minor a. Lines 1144-1145. Solis et al is incorrectly cited as Mol. Cell 2018. The year, volume and page numbers are incorrect.

We have fixed this error.

7) Details of the RT-qPCR assay are more appropriate for the M&Ms, not figure legends.

We agree and made appropriate changes to the figure legend.

8) Line 201: "arginine at the -3 position" is inconsistent with the sequences presented in Fig. 1b.

Our apologies for this error. The PKC phosphorylation substrate motif is typically (R/K)XpS/TX(R/K), where the basic amino acids (Arg or Lys) are at positions -2 and +2 relative to the phosphorylated residue. We have revised the text to reflect this with appropriate references.

9) Inaccurate wording, typos, punctuation issues still exist throughout the ms. For example: lines 130, 142, 190, 310, 905, 913, 915, 920.

We have fixed the wording, typos, punctuation issues throughout the manuscript.